# SQ Lower Bounds for Learning Single Neurons with Massart Noise

**Ilias Diakonikolas**
University of Wisconsin-Madison
ilias@cs.wisc.edu

**Daniel M. Kane**
University of California, San Diego
dakane@cs.ucsd.edu

**Lisheng Ren**
University of Wisconsin-Madison
lren29@wisc.edu

**Yuxin Sun**
University of Wisconsin-Madison
yxsun@cs.wisc.edu

## Abstract

We study the problem of PAC learning a single neuron in the presence of Massart noise. Specifically, for a known activation function $f : \mathbb{R} \to \mathbb{R}$, the learner is given access to labeled examples $(\mathbf{x}, y) \in \mathbb{R}^d \times \mathbb{R}$, where the marginal distribution of $\mathbf{x}$ is arbitrary and the corresponding label $y$ is a Massart corruption of $f(\langle \mathbf{w}, \mathbf{x} \rangle)$. The goal of the learner is to output a hypothesis $h : \mathbb{R}^d \to \mathbb{R}$ with small squared loss. For a range of activation functions, including ReLUs, we establish super-polynomial Statistical Query (SQ) lower bounds for this learning problem. In more detail, we prove that no efficient SQ algorithm can approximate the optimal error within any constant factor. Our main technical contribution is a novel SQ-hard construction for learning $\{\pm 1\}$-weight Massart halfspaces on the Boolean hypercube that is interesting on its own right.

## 1 Introduction

The success of deep learning has served as a motivation for understanding the complexity of learning simple classes of neural networks. Here we study arguably the simplest possible setting of learning a *single* neuron, i.e., a real-valued function of the form $\mathbf{x} \mapsto f(\langle \mathbf{w}, \mathbf{x} \rangle)$, where $\mathbf{w}$ is the weight vector of parameters and $f : \mathbb{R} \mapsto \mathbb{R}$ is a non-linear and monotone activation. The underlying learning problem is the following: Given i.i.d. samples from a distribution $\mathcal{D}$ on $(\mathbf{x}, y)$, where $\mathbf{x} \in \mathbb{R}^d$ is the example and $y \in \mathbb{R}$ is the corresponding label, the goal is to learn the target function in $L_2^2$-loss. That is, the objective of the learner is to output a hypothesis $h : \mathbb{R}^d \mapsto \mathbb{R}$ such that $\mathbf{E}_{(\mathbf{x},y) \sim \mathcal{D}}[(h(\mathbf{x}) - y)^2]$ is as small as possible, compared to the optimal loss value $\mathrm{OPT} := \min_{\mathbf{w} \in \mathbb{R}^d} \mathbf{E}_{(\mathbf{x},y) \sim \mathcal{D}}[(f(\langle \mathbf{w}, \mathbf{x} \rangle) - y)^2]$. A learning algorithm in this context is called proper if the hypothesis $h$ is restricted to be of the form $h_{\widehat{\mathbf{w}}}(\mathbf{x}) = f(\langle \widehat{\mathbf{w}}, \mathbf{x} \rangle)$, for some $\widehat{\mathbf{w}} \in \mathbb{R}^d$. One of the most popular activations is the ReLU function, corresponding to $f(u) = \mathrm{ReLU}(u) \stackrel{\text{def}}{=} \max\{0, u\}$. In this work, we study the complexity of improperly learning single neurons, where the marginal distribution on examples is fixed but arbitrary and the hypothesis $h$ is allowed to be any efficiently computable function.

In the realizable case, i.e., when the labels are consistent with a function in the target concept class, the above learning problem is known to be efficiently solvable for various activation functions. A line of work, see, e.g., [KS09, Sol17, YS20] and references therein, has shown that simple algorithms like gradient-descent efficiently converge to an optimal solution (in some cases under assumptions on the marginal distribution on examples). On the other hand, in the adversarial label noise (aka agnostic) model, known hardness results [Dan16, DKMR22a] rule out efficient constant factor approximations to the optimal loss for a range of activations including ReLUs. The aforementioned negative results

for label agnostic learning motivate the study of weaker corruption models, where non-trivial efficient learning algorithms may still be possible. A natural class of such models — that may be more realistic in some practical applications — are semi-random noise models, involving a combination of adversarial choices and random choices.

Here we focus on the *Massart (or bounded) noise* model [MN06], a classical semi-random model first defined in the context of binary classification (see [Slo88] for an equivalent noise model). Intuitively, in the Massart model, an adversary has control over a (uniformly) *random $\eta < 1/2$* fraction of the labels (see Definition 1.1). In the context of binary classification, [DGT19] gave the first non-trivial learning algorithm for halfspaces in this model (see also [CKMY20, DKT21]). Subsequent work [DK22, NT22] provided evidence that the error guarantee of the latter algorithm is essentially best possible in the Statistical Query (SQ) model [Kea98]; and, more recently, under standard cryptographic assumptions [DKMR22b].

To state our results, we formally define the following natural generalization of the Massart model for real-valued functions (see, e.g., [CKMY21, DPT21]).

**Definition 1.1** (Massart Noise Model). Let $\mathcal{G}$ be a concept class of real-valued functions over $\mathbb{R}^d$, $\mathcal{D}_\mathbf{x}$ be a fixed distribution over $\mathbb{R}^d$, and $0 < \eta < 1/2$. Fix an unknown function $g \in \mathcal{G}$. The noiseless distribution $\mathcal{D}$ (corresponding to $g$) is the distribution on labeled examples $(\mathbf{X}, Y)$, supported on $\mathbb{R}^d \times \mathbb{R}$, where $\mathbf{X} \sim \mathcal{D}_\mathbf{x}$ and $Y = g(\mathbf{X})$. An $\eta$-*Massart distribution*, $\mathcal{D}_\eta^{\mathrm{Mass}}$, is a distribution on labeled examples $(\mathbf{X}, Y')$, supported on $\mathbb{R}^d \times \mathbb{R}$, such that for $(\mathbf{X}, Y') \sim \mathcal{D}_\eta^{\mathrm{Mass}}$ we have that (i) $\mathbf{X} \sim \mathcal{D}_\mathbf{x}$, and (ii) for all $\mathbf{x} \in \mathbb{R}^d$ it holds that $\mathbf{Pr}_{(\mathbf{X}, Y') \sim \mathcal{D}_\eta^{\mathrm{Mass}}}[Y' \neq g(\mathbf{X}) \mid \mathbf{X} = \mathbf{x}] \leq \eta$.

Given sample access to the $\eta$-Massart distribution $\mathcal{D}_\eta^{\mathrm{Mass}}$, corresponding to an unknown $g \in \mathcal{G}$, the goal is to output a hypothesis $h : \mathbb{R}^d \mapsto \mathbb{R}$ such that $\mathcal{L}_2(h; \mathcal{D}_\eta^{\mathrm{Mass}}) := \mathbf{E}_{(\mathbf{X}, Y') \sim \mathcal{D}_\eta^{\mathrm{Mass}}}[(Y' - h(\mathbf{X}))^2]$ is small. Let $\mathrm{OPT}_{\mathrm{Mass}} := \inf_{g \in \mathcal{G}} \mathbf{E}_{(\mathbf{X}, Y') \sim \mathcal{D}_\eta^{\mathrm{Mass}}}[(Y' - g(\mathbf{X}))^2]$ denote the optimal squared error. We will say that a learning algorithm is $\alpha$-*approximate* if it outputs an $h : \mathbb{R}^d \mapsto \mathbb{R}$ that with high probability satisfies $\mathcal{L}_2(h; \mathcal{D}_\eta^{\mathrm{Mass}}) \leq \alpha(d) \cdot \mathrm{OPT}_{\mathrm{Mass}}$. We say that a learner is a *constant factor approximation* if $\alpha = O(1)$. We focus on the concept class of single neurons: for an activation $f : \mathbb{R} \mapsto \mathbb{R}$, we will denote by $\mathcal{C}_f \overset{\text{def}}{=} \{c_\mathbf{w} : \mathbb{R}^d \mapsto \mathbb{R} \mid c_\mathbf{w}(\mathbf{x}) = f(\langle \mathbf{w}, \mathbf{x} \rangle), \mathbf{w} \in \mathbb{R}^d\}$.

In Definition 1.1, the Massart adversary corrupts each label independently with probability *at most* $\eta$. Even though this noise model might appear innocuous, the fact that the corruption probability is unknown to the learner makes the design of efficient Massart learners challenging. The Massart model has been extensively studied in the context of binary classification [Slo88, RS94, Slo96, ABHU15, ABHZ16, DGT19, DKTZ20, CKMY20, DKT21] and, more recently, for learning real-valued functions [CKMY21, DPT21].

For the task of PAC learning halfspaces with Massart noise (i.e., neurons corresponding to the sign activation), there is compelling evidence that even approximate learning is computationally hard [DK22, NT22, DKMR22b]. In sharp contrast, our understanding of the possibilities and limitations of Massart learning well-behaved *real-valued* functions (including ReLUs and other Lipschitz monotone activations) remains limited. On the positive side, recent work developed the first efficient learners for linear regression [CKMY21, DPT21] and ReLU regression [DPT21] with Massart noise. We note that the ReLU regression algorithm of [DPT21] requires a certain anti-concentration condition on the distribution $\mathcal{D}_\mathbf{x}$ of examples, which is crucial for its performance guarantees. In fact, without such an assumption, no non-trivial upper bound is known for ReLUs (or other non-linear activations). This discussion prompts the following question:

> *Is there an efficient $O(1)$-approximate learner for* distribution-free *learning of a single neuron with Massart noise?*

For the important case of ReLU activations, [DPT21] conjectured that the distribution-independent PAC learning problem is intractable. As the main contribution of this paper, we provide strong evidence towards this conjecture, by establishing super-polynomial lower bounds in the Statistical Query (SQ) model — a restricted but powerful family of algorithms. Specifically, we show that no efficient SQ algorithm can achieve *any* constant factor approximation. Moreover, our SQ-hardness result is not specific to ReLUs, but generalizes to a broad class of non-linear activation functions.

## 1.1 Our Results

In this work, we give strong evidence that the problem of learning single neurons with Massart noise does not admit any constant factor approximation. Specifically, we show that any efficient SQ algorithm cannot achieve a constant factor approximation. In fact, the hardness gap that we establish is super-constant, scaling with the dimensionality of the problem.

Instead of directly accessing samples, SQ algorithms [Kea98] are only to adaptively query expectations of bounded functions of the underlying distribution up to some tolerance (see Section 2). The class of SQ algorithms is fairly broad: a wide range of known algorithmic techniques in machine learning are known to be implementable in the SQ model [FGR+17].

For the important class of ReLU activations, our main result is the following:

**Theorem 1.2** (SQ Hardness of Massart Learning ReLUs). *Any SQ algorithm that learns a single neuron with ReLU activation on $\mathbb{R}^d$, in the presence of Massart noise with $\eta = 1/3$, to squared error better than $1/\mathrm{poly}(\log(d))$ requires either queries of accuracy better than $2^{-(\log d)^{c_1}}$ or at least $2^{(\log d)^{c_1}}$ statistical queries, for some constant $c_1 > 1$. This holds even if the optimal squared error is at most $2^{-(\log d)^{c_2}}$ for some $0 < c_2 < 1$, and the total weight of the neuron is $\mathrm{poly}(d)$.*

Theorem 1.2 rules out the existence of efficient SQ algorithms (i.e., using polynomially many queries of inverse polynomial accuracy) with approximation ratio $2^{(\log d)^c}$ for some $0 < c < 1$. It therefore a fortiori rules out any constant factor approximate SQ learner.

We note that the SQ-hardness result of Theorem 1.2 does not require the linearity of the ReLU (on positive inputs); a similar result can be shown for a broader class of activation functions. Specifically, we can generalize our SQ-hardness result to any activation $f$ of the form $f(u) = 0$, $u < 0$, and $\exists u_0 \geq 0, f(u_0) \neq 0$.

Theorem 1.2 establishes SQ-hardness of learning single neurons under the Massart noise notion of Definition 1.1. We note that for learning real-valued functions, one can consider other natural definitions of "Massart noise". Specifically, Definition 1.1 considers an $L_0$-perturbation (the adversary is allowed to arbitrarily corrupt a random $\eta$-fraction of the labels). Another natural definition considers $L_2$-perturbations, as stated below (note that in the definition below, the parameter $\eta$ does not need to be bounded above by $1/2$).

**Definition 1.3** ($L_2$-Massart Noise Model). *Let $\mathcal{G}$ be a concept class of real-valued functions over $\mathbb{R}^d$, $\mathcal{D}_\mathbf{x}$ be a fixed distribution over $\mathbb{R}^d$, and $\eta > 0$. Fix an unknown function $g \in \mathcal{G}$. An $\eta$-$L_2$-Massart distribution, $\mathcal{D}_\eta^{\mathrm{Mass-L2}}$, is a distribution on labeled examples $(\mathbf{X}, Y)$, supported on $\mathbb{R}^d \times \mathbb{R}$, such that for $(\mathbf{X}, Y) \sim \mathcal{D}_\eta^{\mathrm{Mass-L2}}$ we have that (i) $\mathbf{X} \sim \mathcal{D}_\mathbf{x}$, and (ii) for all $\mathbf{x} \in \mathbb{R}^d$ it holds that $\mathbf{E}_{(\mathbf{X},Y)\sim\mathcal{D}_\eta^{\mathrm{Mass-L2}}}[(Y - g(\mathbf{X}))^2 \mid \mathbf{X} = \mathbf{x}] \leq 4\eta$. We will use $\mathrm{OPT}_{\mathrm{Mass-L2}} := \inf_{g \in \mathcal{G}} \mathbf{E}_{(\mathbf{X},Y)\sim\mathcal{D}_\eta^{\mathrm{Mass-L2}}}[(Y - g(\mathbf{X}))^2]$ to denote the optimal squared error.*

Note that for $\{\pm 1\}$ labels, with noise rate $\eta < 1/2$, the above model generalizes the standard Massart model (for binary classification) with the same noise rate $\eta$. For this noise model, we establish SQ-hardness for the following general family of non-linear activations (including ReLUs):

**Definition 1.4** (Fast Convergent Activation). *We say that a function $f : \mathbb{R} \mapsto \mathbb{R}$ is a fast-convergent activation if either $g(t) := f(t)$ or $g(t) := f(-t)$ satisfies the following: (i) $\lim_{t\to-\infty} g(t)$ exists. (ii) For $t < 0$ with absolute value sufficiently large, $|g(t) - g(-\infty)| = 1/\mathrm{poly}(|t|)$.*

Intuitively, the second condition above requires that the function converges to its limit at inverse polynomial rate. Without loss of generality, we consider activation which converge on the negative side. For such an activation $f$, let $f_- := f(-\infty)$ and $c_+$ be a constant such that $f(c_+) \neq f_-$.

Our proof technique establishing Theorem 1.2 is quite robust and can be adapted to $L_2$-Massart noise under fast convergent activations. Our main result in this context is the following:

**Theorem 1.5** (SQ Hardness of $L_2$-Massart Learning). *Let $f : \mathbb{R} \mapsto \mathbb{R}$ be a fast convergent activation. Any SQ algorithm that learns a single neuron with activation $f$ on $\mathbb{R}^d$, in the presence of $\eta$-$L_2$-Massart noise with $\eta = \frac{2(f(c_+)-f_-)^2}{9}$, to squared error better than $1/\mathrm{poly}(\log(d))$ requires either queries of accuracy better than $2^{-(\log d)^{c_1}}$ or at least $2^{(\log d)^{c_1}}$ statistical queries, for some constant $c_1 > 1$. This holds even if the optimal squared error is at most $2^{-(\log d)^{c_2}}$ for some $0 < c_2 < 1$, and the total weight of the neuron is $\mathrm{poly}(d)$.*

Interestingly, the key ingredient for our aforementioned SQ-hardness results for real-valued functions is a new SQ-hardness construction for low-weight halfspaces (i.e., neurons with a sign activation) on the Boolean hypercube. In this context, we prove:

**Theorem 1.6** (SQ Hardness for Low-weight Massart Halfspaces on $\{0,1\}^d$). *Any SQ algorithm that learns $\{\pm 1\}$-weight halfspaces on $\{0,1\}^d$, in the presence of Massart noise with $\eta = 1/3$, to 0-1 error better than $1/\mathrm{poly}(\log(d))$ requires either queries of accuracy better than $2^{-(\log d)^{c_1}}$ or at least $2^{(\log d)^{c_1}}$ statistical queries, for some constant $c_1 > 1$. This holds even if the optimal 0-1 error is at most $2^{-(\log d)^{c_2}}$ for some $0 < c_2 < 1$.*

Theorem 1.6 rules out any efficient polynomial (relative) approximation for $\{\pm 1\}$-weight halfspaces on the hypercube. This is the first hardness result for approximate learning of Boolean Massart halfspaces. Prior work either obtained SQ-hardness of exact learning [CKMY20] or was inherently applicable to halfspaces on $\mathbb{R}^d$ [DK22, NT22].

A number of learning problems involving halfspaces are computationally easy when the weights are small integers (aka in the "large margin" case) and computationally hard for arbitrary weights. The conceptual message of Theorem 1.6 is that the Massart halfspace learning problem is SQ-hard due to the combinatorial nature of the problem (and not due to the magnitude of the weights). This addresses an open problem of [Blu03] regarding the complexity of Massart learning simple halfspaces.

## 1.2 Technical Overview

We start by describing the proof of Theorem 1.6. Our SQ lower bound for learning Boolean Massart halfspaces requires a number of novel ideas. Our starting point is the construction of [DK22] that proves a similar lower bound in the continuous setting. They begin by producing a one-dimensional construction of a Massart PTF whose distributions conditional on $y = 1$ and $y = -1$ approximately match many moments. Using techniques from [DKS17], they show that by embedding this one-dimensional construction into higher dimensions, they can produce $d$-dimensional instances of Massart polynomial threshold functions (PTFs) that are SQ-hard to learn. They then further embed these instances via the Veronese embedding to produce SQ-hard LTF instances (essentially using the fact that a PTF in $x$ is an LTF in the low-degree monomials of $x$).

Our proof adapts this general idea to the discrete setting. The first obstacle is developing an appropriate analogue of the one-dimensional construction. The construction from [DK22] uses the fact that a discrete Gaussian nearly matches moments with a standard Gaussian; thus, making the conditional distributions of $x$ mixtures of discrete Gaussians ensures that the moment matching condition is satisfied. By carefully picking this mixture, they ensure that the conditional distributions have no overlap for $|x|$ small (thus ensuring a small value of OPT), but that the $y = 1$ case is always more likely for $|x|$ sufficiently large. This construction does not work in our case, as we need our one-dimensional construction to be discrete.

Our basic idea is to begin by noting that the binomial distribution conditioned on $x$ being $0$ $\mod s$ approximately matches many moments with the full binomial. As a first attempt, we let $y = -1$ if $x \equiv 0 \mod s$ and $y = 1$ otherwise. This matches many moments with the binomial, but alternates between $y = 1$ and $y = -1$ many times, and thus cannot be considered to be a low-degree PTF with Massart noise. To fix this, we need to modify our distributions so that: (i) Conditioned on any $x$ far from $n/2$, $y$ is more likely to be 1 than $-1$, (ii) the two distributions conditioning on $y = 1$ and $y = -1$ have little overlap, and (iii) each conditional distribution approximately matches moments with the full binomial. We can fix (i) at the cost of (ii) by replacing the conditional distribution on $y = 1$ with the full binomial distribution. As long as the prior probability of $y = 1$ exceeds that of $y = -1$ by enough, even for $x \equiv 0 \mod s$, $y = 1$ will be more likely than $y = -1$. Unfortunately, the conditional distributions now have too much overlap. We can address this by moving the mass in the $y = 1$ conditional off of the points with $x \equiv 0 \mod s$ and $|x - n/2|$ small. Importantly, we must find a way to do this without destroying property (iii). To that end, we show that there is a way to move mass from each of these points $x$ and redistribute it to nearby points in such a way so as to not affect any of the low-order moments (see Lemma 3.8). By doing this to each $x \equiv 0 \mod s$ with $|x - n/2|$ small, we get our final construction.

We also need to modify the method by which we embed the one-dimensional construction into higher dimensions in order to obtain the family of SQ-hard PTF instances. This construction must differ from previous constructions, as our family of distributions will be discrete and not Gaussian-

like as in [DKS17]. Fortunately, we can leverage the recent technique of [DKS22], embedding our low-dimensional construction as a junta. In particular, a significant difference with the Gaussian case is in the way we embed the low-dimensional distribution as a higher-dimensional one. In the Gaussian case, we simply take the distribution to be Gaussian in independent directions. In our discrete setting, we begin by embedding into a moderate dimensional hypercube by taking the unique symmetric distribution, where our one-dimensional distribution over some subset $S$ is the distribution over $\sum_{i \in S} X_i$. We note that this distribution will approximately match low-degree moments with the uniform distribution over the hypercube. We then embed this distribution into a higher-dimensional hypercube as a random junta.

As an application of the above general recipe to obtain SQ lower bounds for discrete distributions, we note that the hard instances we construct for learning Boolean halfspaces with Massart noise, *also* (with a slight change of variables) produce hard instances for ReLUs (and other activations). In particular, in our hard instance for PTFs, the optimal classifier $f$ is given by $f(\mathbf{x}) = -1$ if $\mathbf{x}_S \equiv 0 \mod s$ and $|\mathbf{x}_S - n/2| < ds/2$, and 1 otherwise, where $\mathbf{x}_S$ is the sum over the coordinates of $\mathbf{x}$ in some particular subset $S$. We note that the function $(1 - f(\mathbf{x}))/2$, which is equal to 1 if $\mathbf{x}_S \equiv 0 \mod s$ and $|\mathbf{x}_S - n/2| < ds/2$ and 0 otherwise, can be written as $\mathrm{ReLU}(p(\mathbf{x}))$ for some degree $O(d)$ polynomial $p$, where $p(\mathbf{x}) = 1$ for $\mathbf{x}_S \equiv 0 \mod s$ and $|\mathbf{x}_S - n/2| < ds/2$, and $p(\mathbf{x}) \leq 0$ otherwise. By replacing $\mathbf{x}$ by its Vernonese embedding as before, we can produce hard instances of ReLU functions with Massart noise.

## 2 Preliminaries

**Notation** For $n \in \mathbb{Z}_+$, we denote $[n] \stackrel{\text{def}}{=} \{1, \ldots, n\}$. For two distributions $p, q$ over a probability space $\Omega$, let $d_{\mathrm{TV}}(p, q) = \sup_{S \subseteq \Omega} |p(S) - q(S)|$ denote the total variation distance between $p$ and $q$. We use $\mathbf{Pr}[\mathcal{E}]$ and $\mathbb{I}[\mathcal{E}]$ for the probability and the indicator of event $\mathcal{E}$. For a real random variable $X$, we use $\mathbf{E}[X], \mathbf{Var}[X]$ to denote the expectation and variance of $X$, respectively. For $n \in \mathbb{Z}_+$ and $0 \leq p \leq 1$, we use $\mathrm{Bin}(n, p)$ to denote the Binomial distribution with parameters $n$ and $p$. Throughout this article, we will use capital letters (e.g., $X, \mathbf{X}$) to denote random variables and random vectors, and small letters (e.g, $x, \mathbf{x}$) to denote corresponding values.

**Statistical Query Algorithms** We will use the framework of Statistical Query (SQ) algorithms for problems over distributions [FGR+17]. We require the following standard definition.

**Definition 2.1** (Decision/Testing Problem over Distributions). Let $D$ be a distribution and $\mathcal{D}$ be a family of distributions over $\mathbb{R}^M$. We denote by $\mathcal{B}(\mathcal{D}, D)$ the decision (or hypothesis testing) problem in which the input distribution $D'$ is promised to satisfy either (a) $D' = D$ or (b) $D' \in \mathcal{D}$, and the goal of the algorithm is to distinguish between these two cases.

We define SQ algorithms as algorithms that do not have direct access to samples from the distribution, but instead have access to an SQ oracle. We will consider the following standard oracle.

**Definition 2.2** (STAT Oracle). Let $D$ be a distribution on $\mathbb{R}^M$. A *Statistical Query (SQ)* is a bounded function $f : \mathbb{R}^M \to [-1, 1]$. For $\tau > 0$, the $\mathrm{STAT}(\tau)$ oracle responds to the query $f$ with a value $v$ such that $|v - \mathbf{E}_{X \sim D}[f(X)]| \leq \tau$. We call $\tau$ the *tolerance* of the statistical query. A *Statistical Query (SQ) algorithm* is an algorithm whose objective is to learn some information about an unknown distribution $D$ by making adaptive calls to the corresponding $\mathrm{STAT}(\tau)$ oracle.

## 3 SQ Hardness Construction for Supervised Learning

### 3.1 Generic SQ Lower Bound Construction

We start with some basic definitions. Let $U_M$ be the uniform distribution over $\{0, 1\}^M$. For a subset $T \subseteq [M]$ and $\mathbf{x} \in \{0, 1\}^M$, we denote $\chi_T(\mathbf{x}) = (-1)^{\sum_{i \in T} x_i}$. For a distribution $\mathbf{P}$ over $\{0, 1\}^M$, let $\widehat{\mathbf{P}}(T) = \mathbf{E}_{\mathbf{X} \sim \mathbf{P}}[\chi_T(\mathbf{X})]$. We will require the orthogonal polynomials under the binomial distribution.

**Definition 3.1** (Kravchuk Polynomial [Sze89]). For $k, m, x \in \mathbb{Z}_+$ with $0 \leq k, x \leq m$, the Kravchuk polynomial $\mathcal{K}_k(x; m)$ is the univariate degree-$k$ polynomial in $x$ defined by $\mathcal{K}_k(x; m) := \sum_{T \subseteq [m], |T| = k} \chi_T(\mathbf{y}) = \sum_{j=0}^{k} (-1)^j \binom{x}{j} \binom{m-x}{k-j}$, where $\mathbf{y}$ has $x$ 1's and $m - x$ 0's.

The following distribution family that is the basis of our discrete SQ lower bound construction.

**Definition 3.2** (High-Dimensional Hidden Junta Distribution). Let $m, M \in \mathbb{Z}_+$ with $m < M$. For a distribution $A$ on $[m] \cup \{0\}$ with probability mass function (pmf) $A(x)$ and a subset $S \subseteq [M]$ with $|S| = m$, consider the probability distribution over $\{0, 1\}^M$, denoted by $\mathbf{P}_S^A$, such that for $\mathbf{X} \sim \mathbf{P}_S^A$ the distribution $(X_i)_{i \notin S}$ is the uniform distribution on its support and the distribution $(X_i)_{i \in S}$ is symmetric with $\sum_{i \in S} X_i$ distributed according to $A$. Specifically, $\mathbf{P}_S^A$ is given by the pmf $\mathbf{P}_S^A(\mathbf{x}) = 2^{-M+m} A\left(\sum_{i \in S} x_i\right) \binom{m}{\sum_{i \in S} x_i}^{-1}$.

The following condition describes the approximate moment-matching property of the desired distribution $A$ with the Binomial distribution.

**Condition 3.3.** *Let $k, m \in \mathbb{Z}_+$ with $k < m$ and $\nu > 0$. The distribution $A$ on $[m] \cup \{0\}$ is such that $|\mathbf{E}_{X \sim A}[\mathcal{K}_t(X; m)]| \leq \nu$, for all $1 \leq t \leq k$.*

We now define the hypothesis testing and learning problem which will be used throughout this paper:

**Definition 3.4** (Hidden Junta Binary Testing Problem). Fix $a \neq b \in \mathbb{R}$. Let $A$ and $B$ be distributions on $[m] \cup \{0\}$ satisfying Condition 3.3 with parameters $k \in \mathbb{Z}_+$ and $\nu \in \mathbb{R}_+$, and let $p \in (0, 1)$. For $M \in \mathbb{Z}_+$, $M > m$, and a subset $S \subseteq [M]$ with $|S| = m$, define the distribution $\mathbf{P}_{S,a,b}^{A,B,p}$ on $\{0, 1\}^M \times \{a, b\}$ that returns a sample from $(\mathbf{P}_S^A, a)$ with probability $p$ and a sample from $(\mathbf{P}_S^B, b)$ with probability $1 - p$. In the $(A, B, a, b, M)$-Hidden Junta Testing Problem, one is given access to a distribution $D$ so that either $H_0$: $D = U_M^p$, where for $(\mathbf{X}, Y) \sim U_M^p$ we have that $\mathbf{X}$ is a uniform random element of $\{0, 1\}^M$, and $Y$ is independently $a$ with probability $p$ and $b$ with probability $1 - p$. $H_1$: $D$ is given by $\mathbf{P}_{S,a,b}^{A,B,p}$ for some subset $S \subseteq [M]$ with $|S| = m$. One is then asked to distinguish between $H_0$ and $H_1$.

Note that this is just the hypothesis testing problem $\mathcal{B}(\mathcal{D}, D)$ with $D = U_M^p$ and $\mathcal{D} = \{\mathbf{P}_{S,a,b}^{A,B,p}\}$.

**Proposition 3.5** (Generic Discrete SQ Lower Bound). *Let $m, M \in \mathbb{Z}_+$ with $M > m$. Let $A, B$ be distributions on $[m] \cup \{0\}$ satisfying Condition 3.3. Let $\tau \geq k\nu^2 + 2^{-k}(\chi^2(A, \text{Bin}(m, 1/2)) + \chi^2(B, \text{Bin}(m, 1/2)))$. Any SQ algorithm that solves the testing problem of Definition 3.4 with probability at least 2/3 must either make queries of accuracy better than $\sqrt{2\tau}$ or must make at least $2^{\Omega(m)}\tau/(\chi^2(A, \text{Bin}(m, 1/2)) + \chi^2(B, \text{Bin}(m, 1/2)))$ statistical queries.*

We give a proof sketch here and defer the full proof to Appendix B. We first pick a collection $\mathcal{C}$ of $s = 2^{\Omega(m)}$ subsets $S \subseteq [M]$ of size $m$ whose pairwise intersections are small. For any $S, S' \in \mathcal{C}$, we write the correlation between $\mathbf{P}_{S,a,b}^{A,B,p}, \mathbf{P}_{S',a,b}^{A,B,p}$ as a linear combination of the correlation between $\mathbf{P}_S^A, \mathbf{P}_{S'}^A$ and the correlation between $\mathbf{P}_S^B, \mathbf{P}_{S'}^B$. Since we can directly obtain an upper bound for the correlation between the hidden junta distributions, applying existing techniques will yield our result.

## 3.2 Construction of Univariate Moment-Matching Distributions

Here we give the construction of our approximate moment-matching distributions. For convenience, we use the "expectation" and "moments" for the unnormalized measure without clarification. The main result of this section is captured in the following proposition.

**Proposition 3.6.** *Let $d, k, s, m \in \mathbb{Z}_+$ and $\zeta \in (0, 1/2)$ such that: (i) $s \geq \omega(k^4)$, (ii) $k < m/2$, (iii) $ds \geq \Omega(\sqrt{m \log(1/\zeta)})$, and (iv) $s^2 d \leq o(m)$. There exist measures $\mathcal{D}_+$ and $\mathcal{D}_-$ over $[m] \cup \{0\}$ and a union $J$ of $d$ points in $[m] \cup \{0\}$ such that: 1. (a) $\mathcal{D}_+ = 0$ on $J$, and (b) $\mathcal{D}_+ > 2\mathcal{D}_-$ on $\overline{J} = [m] \cup \{0\} \setminus J$. 2. All but $\zeta$-fraction of the measure of $\mathcal{D}_-$ lies in $J$. 3. The distributions $\mathcal{D}_+/\|\mathcal{D}_+\|_1$ and $\mathcal{D}_-/\|\mathcal{D}_-\|_1$ satisfy Condition 3.3 with parameters $k$ and $\nu \leq \binom{m}{k} \exp(-\Omega(m/s^2))$. 4. (a) $\mathcal{D}_+$ is at most $O(1)\text{Bin}(m, 1/2)$ and (b) $\|\mathcal{D}_+\|_1 = \Theta(1)$. 5. $\|D_-\|_1 = \Theta(1/s)$.*

*Proof.* We start by constructing each measure in turn.

**Definition of the Measure $\mathcal{D}_-$.** We define the measure $\mathcal{D}_-$ as follows: $\mathcal{D}_-(x) := \text{Bin}(m, 1/2)(x)$ if $x \equiv 0 \pmod{s}$; otherwise $\mathcal{D}_-(x) = 0$. We claim that this satisfies Condition 3.3. This is shown in the following lemma.

**Lemma 3.7.** $\mathcal{D}_-(x)$ *satisfies Condition 3.3 with parameters $k$ and $\nu = s\binom{m}{k} \exp(-\Omega(m/s^2))$.*

*Proof.* We need to bound $\mathbf{E}_{Z\sim\mathcal{D}_-}[\mathcal{K}_t(Z;m)]$ for $1 \le t \le k$. By definition, we have that

$$\mathbf{E}_{Z\sim\mathcal{D}_-}[\mathcal{K}_t(Z;m)] = \sum_{T\subseteq[m],|T|=t} \mathbf{E}_{Z\sim\mathcal{D}_-}[\chi_T(\mathbf{Y})] = \binom{m}{t}\mathbf{E}_{\mathbf{X}\sim\mathcal{R}}[\chi_{T_0}(\mathbf{X})],$$

where $\mathbf{Y}$ has $Z$ 1's and $m - Z$ 0's, and $\mathcal{R} \in \{0,1\}^m$ is the unique symmetric measure with $\sum_{i=1}^m X_i$ having measure $\mathcal{D}_-$, and $T_0 \subseteq [m]$ is some subset with $|T_0| = t$. Let $\omega$ be a primitive $s^{th}$ root of unity. We note that the pmf $\mathcal{R}(\mathbf{x})$ of the measure $\mathcal{R}$ satisfies

$$\mathcal{R}(\mathbf{x}) = \frac{1}{2^m s}\sum_{j=0}^{s-1}\omega^{\left(j\sum_{i=1}^m x_i\right)} = \frac{1}{2^m s}\sum_{j=0}^{s-1}\prod_{i=1}^m \omega^{jx_i}.$$

Therefore, we can write

$$\mathcal{R}(\mathbf{x})\chi_{T_0}(\mathbf{x}) = \left((-1)^{\sum_{i=1}^m \mathbb{I}[i\in T_0]x_i}\right)\left(\frac{1}{2^m s}\sum_{j=0}^{s-1}\prod_{i=1}^m \omega^{jx_i}\right) = \frac{1}{2^m s}\sum_{j=0}^{s-1}\prod_{i=1}^m \left(\omega^j(-1)^{\mathbb{I}[i\in T_0]}\right)^{x_i}.$$

Since the expectation is the sum of the above over all $x \in \{0,1\}^m$ and since this separates as a product, we get that

$$\mathbf{E}_{\mathbf{X}\sim\mathcal{R}}[\chi_{T_0}(\mathbf{X})] = \frac{1}{2^m s}\sum_{\mathbf{x}\in\{0,1\}^m}\sum_{j=0}^{s-1}\prod_{i=1}^m \left(\omega^j(-1)^{\mathbb{I}[i\in T_0]}\right)^{x_i} = \frac{1}{2^m s}\sum_{j=0}^{s-1}\prod_{i=1}^m \left(1+\omega^j(-1)^{\mathbb{I}[i\in T_0]}\right).$$

Note that the terms with $2j \equiv 0 \pmod{s}$ have indices $i$ such that $\omega^j(-1)^{\mathbb{I}[i\in T_0]} = -1$, and do not contribute to the sum. Other terms will have each value of $|1+\omega^j(-1)^{\mathbb{I}[i\in T_0]}|$ at most $2 - \Omega(1/s^2)$. Therefore, $\mathbf{E}_{\mathbf{X}\sim\mathcal{R}}[\chi_{T_0}(\mathbf{X})] = \exp(-\Omega(m/s^2))$. This completes our proof. $\qquad\square$

We also note that $\mathcal{D}_-$ is clearly bounded above by $\mathrm{Bin}(m,1/2)$. We define $J$ to be the union of the $d$ elements of $m \cup \{0\}$ congruent to 0 modulo $s$ that are closest to $m/2$. We note that the measure of $\mathcal{D}_-$ outside $J$ is clearly at most the probability that $\mathrm{Bin}(m,1/2)$ is more than $ds/2$ from $m/2$, which is at most $\zeta$ by standard tail bounds.

**Definition of the Measure $\mathcal{D}_+$.** Intuitively, we would like to define $\mathcal{D}_+$ to be equal to some suitable multiple (say, 3) of the standard Binomial measure $\mathrm{Bin}(m,1/2)$. Such a definition would satisfy the desired moment-matching conditions (property 3 of Proposition 3.6) with zero error and would also guarantee that $\mathcal{D}_+ > 2\mathcal{D}_-$ on $\overline{J}$, as desired (property 1(b)). However, this candidate definition does not satisfy property 1(a), i.e., that $\mathcal{D}_+$ be equal to 0 on $J$. To satisfy the latter property, we will need to carefully modify this measure. The key lemma is the following (see Appendix B for the proof):

**Lemma 3.8.** *Let $s \ge \omega(k^4)$. There exists a signed measure $\mu$ on $\{-s+1, -s+2, \ldots, s-1\}$ such that: (i) For any integer $0 \le t \le k$, $\sum_{i=1-s}^{s-1}\mu(i)i^t = 0$, (ii) $\mu(0) = -1$, (iii) $|\mu(i)| < 1/10, i \ne 0$.*

We are now ready to construct the measure $\mathcal{D}_+$. We begin with the measure $3\mathrm{Bin}(m,1/2)$. We then for each element $x \in J$ take the measure $\mu$ from Lemma 3.8, translate it to center around $x$ and add an appropriate multiple of it to $\mathcal{D}_+$ in order to make $\mathcal{D}_+(x) = 0$. It is clear that the first $k$ moments of $\mathcal{D}_+$ agree with those moments of $3\mathrm{Bin}(m,1/2)$, and from there it follows that $\mathcal{D}_+$ satisfies Condition 3.3 with $\nu = 0$, since for any $0 \le t \le k$ and any point $x \in J$, we have that

$$\sum_{i=1-s}^{s-1}\mu(i)(x+i)^t = \sum_{i=1-s}^{s-1}\mu(i)\sum_{\ell=0}^t \binom{t}{\ell}i^\ell x^{t-\ell} = \sum_{\ell=0}^t \binom{t}{\ell}x^{t-\ell}\sum_{i=1-s}^{s-1}\mu(i)i^\ell = 0,$$

which means that we never change the moments by making $\mathcal{D}_+(x) = 0$. Therefore, we have $\mathcal{D}_+$ is 0 on $J$ by our construction. We also claim that $\mathcal{D}_+$ is bounded between $2\mathrm{Bin}(m,1/2)$ and $4\mathrm{Bin}(m,1/2)$ on $\overline{J}$. For this, we note that for any $x \notin J$, there are at most two integers, $x'$ and $x''$, that are in $J$ and within distance $s$ of $x$. It is clear that

$$|\mathcal{D}_+(x) - 3\mathrm{Bin}(m,1/2)(x)| \le (3/10)(\mathrm{Bin}(m,1/2)(x') + \mathrm{Bin}(m,1/2)(x'')).$$

It suffices to show that $\frac{\mathrm{Bin}(m,1/2)(x')}{\mathrm{Bin}(m,1/2)(x)} < 3/2$ along with the analogous statement for $x''$. However, the log of the ratio is easily seen to be $O(s^2 d/m) = o(1)$, which suffices. This completes the proof of Proposition 3.6. $\qquad\square$

### 3.3 Parameter Setting for the SQ-hard Distributions

We will consider the following family of hardness distributions which will be used in the proof of all SQ hardness results throughout this article. Let $C > 0$ be a sufficiently large universal constant. Let $m$ be a positive integer and $m'$ be an integer on the order of $Cm$. Let $d$ be an integer on the order of $m^{1/10}$, $s$ an integer on the order of $m^{4/9}$, and $k$ an integer on the order of $m^{2/19}$. Observe that $\binom{2d+m'}{m'} \le (m')^{2d} = \exp(O(Cm^{1/10}\log(m)))$. Select $m$ as large as possible so that the above is less than $M$. Decreasing $M$ if necessary, we can assume that $M = \binom{2d+m'}{m'}$. We consider the Veronese mapping $V_{O(d)} : \mathbb{R}^{m'} \to \mathbb{R}^M$, such that the coordinate functions of $V_{O(d)}$ are exactly the monomials in $m'$ variables of degree at most $O(d)$. We define measures $\mathcal{D}_+$ and $\mathcal{D}_-$ on $[m] \cup \{0\}$, as given by Proposition 3.6, with $k, s$ and $d$ as above, and taking $\log(1/\zeta)$ a sufficiently small multiple of $(ds)^2/m$, so that $\zeta = \exp(-\Omega(m^{4/45})) = \exp(-\Omega(\log(M)^{8/9}))$. It is easily verified that these parameters satisfy the assumptions of Proposition 3.6. For a subset $S \subseteq [m']$ of size $m$ and labels $a \ne b \in \mathbb{R}$, define the distribution $\mathbf{P}_{S,a,b}^{\mathcal{D}_+,\mathcal{D}_-,p}$ as in Definition 3.4, with $p = \|\mathcal{D}_+\|_1/(\|\mathcal{D}_+\|_1 + \|\mathcal{D}_-\|_1)$. We will consider the distribution $(\mathbf{X}', Y')$ on $\{0,1\}^M \times \{a,b\}$ by drawing $(\mathbf{X}, Y)$ from $\mathbf{P}_{S,a,b}^{\mathcal{D}_+,\mathcal{D}_-,p}$ and letting $\mathbf{X}' = V_{O(d)}(\mathbf{X})$ and $Y' = Y$. It is easy to see that finding a hypothesis that predicts $y'$ given $\mathbf{x}'$ is equivalent to finding a hypothesis for $y$ given $\mathbf{x}$ (since $y = y'$ and there is a known 1-1 mapping between $\mathbf{x}$ and $\mathbf{x}'$). The pointwise bounds on $\mathcal{D}_+$ and $\mathcal{D}_-$ imply that $\chi^2(\mathcal{D}_+/\|\mathcal{D}_+\|_1, \mathrm{Bin}(m,1/2)) + \chi^2(\mathcal{D}_-/\|\mathcal{D}_-\|_1, \mathrm{Bin}(m,1/2)) = O(s^2)$. The parameter $\nu$ in Proposition 3.5 is at most $sm^k \exp(-\Omega(m/s^2)) = \exp(-\Omega(m^{1/9}))$. Note that as $M = \exp(\widetilde{O}(m^{1/10}))$, this is $\exp(-\Omega(\log(M)^{1.1}))$. As $k$ is also $\Omega(\log(M)^{1.05})$, we have that $\tau = \exp(-\Omega(\log(M)^{1.05})) \le 1/\mathrm{poly}(M)$. In the remaining part of this article, we will use $\mathbf{x}', \mathbf{X}', y', Y'$ without clarification to denote the results of $\mathbf{x}, \mathbf{X}, y, Y$ after the Veronese mapping $V_{O(d)} : \mathbb{R}^{m'} \to \mathbb{R}^M$.

## 4 Concrete SQ Hardness Results

In this section, we prove our SQ hardness results for Massart learning low-weight half-spaces and ReLUs. We provide additional SQ hardness results for learning fast convergent activations with respect to $L_2$-Massart noise in Appendix D.

### 4.1 SQ Hardness of Learning Low-Weight Boolean Halfspaces with Massart Noise

In this subsection, we prove the following theorem.

**Theorem 4.1** (SQ Hardness for Low-weight Massart Halfspaces on $\{0,1\}^M$)**.** *Any SQ algorithm that learns $\{\pm 1\}$-weight halfspaces on $\{0,1\}^M$, in the presence of Massart noise with $\eta = 1/3$, to 0-1 error better than $1/\mathrm{poly}(\log(M))$ requires either queries of accuracy better than $\tau := \exp(-\Omega(\log(M)^{1.05}))$ or at least $1/\tau$ statistical queries. This holds even if the optimal classifier has 0-1 error $\exp(-\Omega(\log M)^{8/9})$.*

*Proof.* Our proof will make use of the SQ framework of Section 3.1 and will crucially rely on the one-dimensional construction of Proposition 3.6. In this subsection, we fix the labels $a = 1, b = -1$, and apply the construction in Section 3.3 to obtain the joint distributions $(\mathbf{X}, Y)$ and $(\mathbf{X}', Y')$. Note that $y = y'$ and there is a known 1-1 mapping between $\mathbf{x}$ and $\mathbf{x}'$, therefore finding a hypothesis that predicts $y'$ given $\mathbf{x}'$ is equivalent to finding a hypothesis for $y$ given $\mathbf{x}$.

**Claim 4.2.** *The distribution $(\mathbf{X}', Y')$ over $\{0,1\}^M \times \{\pm 1\}$ is a Massart LTF distribution with optimal misclassification error $\mathrm{OPT}_{\mathrm{Mass}} \le \exp(-\Omega(\log(M)^{8/9}))$ and Massart noise rate upper bound of $\eta = 1/3$.*

We defer the proof of the above claim to Appendix C. We now show that the $(\mathcal{D}_+, \mathcal{D}_-, 1, -1, m')$-Hidden Junta Testing Problem efficiently reduces to our learning task. In more detail, we show that any SQ algorithm that computes a hypothesis $h'$ satisfying $\mathbf{Pr}_{(\mathbf{X}',Y')}[h'(\mathbf{X}') \ne Y'] < \min(p, 1-p) - 2\sqrt{2\tau}$ can be used as a black-box to distinguish between $\mathbf{P}_{S,a,b}^{\mathcal{D}_+,\mathcal{D}_-,p}$, for some unknown subset $S \subseteq [m]$ with $|S| = m$, and $U_{m'}^p$. Since there is a 1-1 mapping between $\mathbf{x} \in \{0,1\}^{m'}$ and $\mathbf{x}' \in \{0,1\}^M$, we denote $h : \{0,1\}^{m'} \mapsto \{\pm 1\}$ to be $h(\mathbf{x}) = h'(\mathbf{x}')$. We note that we

can (with one additional query to estimate the $\mathbf{Pr}[h'(\mathbf{X}') \neq Y']$ within error $\sqrt{2\tau}$) distinguish between (i) the distribution $\mathbf{P}_{S,a,b}^{\mathcal{D}_+,\mathcal{D}_-,p}$, and (ii) the distribution $U_{m'}^p$. This is because for any $h$ we have that $\mathbf{Pr}_{(\mathbf{X},Y)\sim U_{m'}^p}[h(\mathbf{X}) \neq Y] \geq \min(p, 1-p)$. Applying Proposition 3.5, we determine that any SQ algorithm which, given access to a distribution $\mathbf{P}$ so that either $\mathbf{P} = U_{m'}^p$, or $\mathbf{P}$ is given by $\mathbf{P}_{S,a,b}^{\mathcal{D}_+,\mathcal{D}_-,p}$ for some unknown subset $S \subseteq [m']$ with $|S| = m$, correctly distinguishes between these two cases with probability at least $2/3$ must either make queries of accuracy better than $\sqrt{2\tau}$ or must make at least $2^{\Omega(m)}\tau/(\chi^2(A, \mathrm{Bin}(m, 1/2)) + \chi^2(B, \mathrm{Bin}(m, 1/2)))$ statistical queries. Therefore, it is impossible for an SQ algorithm to learn a hypothesis with error better than $\min(p, 1-p) - 2\sqrt{2\tau} = \Theta(1/s) - \Theta(\sqrt{\tau}) = 1/\mathrm{polylog}(M)$ without either using queries of accuracy better than $\tau$ or making at least $2^{\Omega(m)}\tau/\mathrm{polylog}(M) > 1/\tau$ many queries. This completes the proof of the SQ-hardness.

It remains to argue that the underlying halfspaces in the hard instance can be assumed to have $\{\pm 1\}$ weights. To deal with the weights, we note that $g$ is a degree-$2d$ PTF that can be defined as the product of $2d$ linear polynomials $L_i$, so that each $L_i$ has integer coefficients and the sum of the absolute values of these coefficients is $O(m)$. This means that $g$ can be defined by a degree-$2d$ polynomial with integer coefficients and the sum of whose absolute values is at most $O(m)^{2d} = \mathrm{poly}(M)$. By doubling these coefficients, we can assume that they are all even. Therefore, the linear threshold function $L$ can be defined by a linear polynomial with even integer weights each of which has absolute value at most $W$. If we replace our distribution over $\{0,1\}^M$ by a distribution over $\{0,1\}^{MW}$ by duplicating each coordinate $W$ times (i.e., creating a new distribution with coordinates $z_{i,j}$ for $i \in [M]$ and $j \in [W]$ with $z_{i,j} = x_i$ for all $i,j$), we can rewrite $L(x)$ as an LTF $L'(z)$, where $L'$ has $\{\pm 1\}$-weights. This is done by replacing a term $a_i x_i$ by $\sum_{j=1}^{(a_i+W)/2} z_{i,j} - \sum_{j=(a_i+W)/2+1}^{W} z_{i,j}$. This completes the proof of Theorem 4.1. $\qquad\square$

## 4.2 SQ Hardness of Learning a Single Neuron with Massart Noise

In this subsection, we prove our SQ hardness result of learning a single neuron with ReLU activation and Massart noise. The standard ReLU function is defined by $\mathrm{ReLU}(t) = \max(t, 0), \forall t \in \mathbb{R}$. For technical convenience, we will consider the following linear transformation of the standard ReLU, $\widehat{\mathrm{ReLU}}(t) = -1$ if $t < 0$, and $\widehat{\mathrm{ReLU}}(t) = -1 + 2t$ otherwise. We note that our SQ hardness result for the $\widehat{\mathrm{ReLU}}$ function applies to the standard ReLU function as well.

**Theorem 4.3** (SQ Hardness of Massart Learning ReLUs). *Any SQ algorithm that learns a single neuron with ReLU activation on $\mathbb{R}^M$, in the presence of Massart noise with $\eta = 1/3$, within squared error better than $1/\mathrm{poly}(\log(M))$ requires either queries of accuracy better than $\tau := \exp(-\Omega(\log(M)^{1.05}))$ or at least $1/\tau$ statistical queries. This holds even if (i) the optimal neuron has squared error $\exp(-\Omega(\log M)^{8/9})$, (ii) The $\mathbf{X}$ values are supported on $\{0,1\}^M$, and (iii) the total weight of the neuron is $\mathrm{poly}(M)$.*

Throughout this subsection, we need the following technical lemma.

**Lemma 4.4.** *Let $J$ be a union of $d$ points in $[m] \cup \{0\}$ for some odd integer $d$. Then there exists a real univariate polynomial $p(x)$ of degree $O(d)$ such that $p(x) = 1, \forall x \in J$, and $p(x) \leq 0, \forall x \in \bar{J}$. In addition, the absolute value of the coefficients of $p(x)$ is at most $m^{O(d)} = \mathrm{poly}(M)$.*

*Proof.* Let $J = \{x_1, \ldots, x_d\}$. Define $q(x) = -\prod_{i=1}^{d}(x - (x_i - 1/2))(x - (x_i + 1/2))$. By definition, we have that $q(x) > 0, \forall x \in J$, and $q(x) < 0, \forall x \in \bar{J}$. Then, by polynomial interpolation, there exists a real univariate polynomial $r$ of degree $d-1$ such that $r(x_i) = \frac{1}{\sqrt{q(x_i)}}, 1 \leq i \leq d$.

Consider the real univariate polynomial $p(x) = r^2(x)q(x)$. For any $1 \leq i \leq d$, we have that $p(x_i) = r^2(x_i)q(x_i) = 1$ and for any $x \in \bar{J}$, we have that $p(x) \leq 0$ since $q(x) < 0, \forall x \in \bar{J}$. Finally by polynomial interpolation, we know that the absolute value of every coefficient of $r(x), p(x)$ is at most $m^{O(d)} = \mathrm{poly}(M)$. $\qquad\square$

*Proof of Theorem 4.3.* Our proof will make use of the SQ framework of Section 3.1 and will crucially rely on the one-dimensional construction of Proposition 3.6. In this section, we fix the labels $a = -1, b = 1$, and apply the construction in Section 3.3 to obtain the joint distributions $(\mathbf{X}, Y)$ and

$(\mathbf{X}', Y')$. Note that $y = y'$ and there is a known 1-1 mapping between $\mathbf{x}$ and $\mathbf{x}'$, therefore finding a hypothesis that predicts $y'$ given $\mathbf{x}'$ is equivalent to finding a hypothesis for $y$ given $\mathbf{x}$.

**Claim 4.5.** *The distribution* $(\mathbf{X}', Y')$ *over* $\{0,1\}^M \times \{\pm 1\}$ *is a Massart single neuron distribution with ReLU activation and with optimal squared error* $\mathrm{OPT}_{\mathrm{Mass-L2}} \le \exp(-\Omega(\log(M)^{8/9}))$ *and Massart noise rate upper bound of* $\eta = 1/3$.

We defer the proof of the above claim to Appendix C. We now show that the $(\mathcal{D}_+, \mathcal{D}_-, -1, 1, m')$-Hidden Junta Testing Problem efficiently reduces to our learning task. In more detail, we show that any SQ algorithm that computes a hypothesis $h'$ satisfying $\mathbf{E}_{(\mathbf{X}',Y')}[(h'(\mathbf{X}') - Y')^2] < 4p - 4p^2 - 2\sqrt{2\tau}$ can be used as a black-box to distinguish between $\mathbf{P}_{S,a,b}^{\mathcal{D}_+,\mathcal{D}_-,p}$, for some unknown subset $S \subseteq [m']$ with $|S| = m$, and $U_{m'}^p$. Since there is a 1-1 mapping between $\mathbf{x} \in \{0,1\}^{m'}$ and $\mathbf{x}' \in \{0,1\}^M$, we denote $h : \{0,1\}^{m'} \mapsto \mathbb{R}$ to be $h(\mathbf{x}) = h'(\mathbf{x}')$. We note that we can (with one additional query to estimate the $\mathbf{E}[(h'(\mathbf{X}') - Y')^2]$ within error $\sqrt{2\tau}$) distinguish between (i) the distribution $\mathbf{P}_{S,a,b}^{\mathcal{D}_+,\mathcal{D}_-,p}$, and (ii) the distribution $U_{m'}^p$. This is because for any $h$ we have that

$$\mathbf{E}_{(\mathbf{X},Y)\sim U_{m'}^p}[(h(\mathbf{X}) - Y)^2] = 1 - 2(1 - 2p)\mathbf{E}_{(\mathbf{X},Y)\sim U_{m'}^p}[h(\mathbf{X})] + \mathbf{E}_{(\mathbf{X},Y)\sim U_{m'}^p}[h(\mathbf{X})^2]$$

$$\ge 1 - 2(1 - 2p)\mathbf{E}_{(\mathbf{X},Y)\sim U_{m'}^p}[h(\mathbf{X})] + \mathbf{E}_{(\mathbf{X},Y)\sim U_{m'}^p}[h(\mathbf{X})]^2 \ge 4p - 4p^2 .$$

Applying Proposition 3.5, we determine that any SQ algorithm which, given access to a distribution $\mathbf{P}$ so that either $\mathbf{P} = U_{m'}^p$, or $\mathbf{P}$ is given by $\mathbf{P}_{S,a,b}^{\mathcal{D}_+,\mathcal{D}_-,p}$ for some unknown subset $S \subseteq [m']$ with $|S| = m$, correctly distinguishes between these two cases with probability at least $2/3$ must either make queries of accuracy better than $\sqrt{2\tau}$ or must make at least $2^{\Omega(m)}\tau/(\chi^2(A, \mathrm{Bin}(m, 1/2)) + \chi^2(B, \mathrm{Bin}(m, 1/2)))$ statistical queries. Therefore, it is impossible for an SQ algorithm to learn a hypothesis with error better than $4p - 4p^2 - 2\sqrt{2\tau} = \Theta(1/s) - \Theta(\sqrt{\tau}) = 1/\mathrm{polylog}(M)$ without either using queries of accuracy better than $\tau$ or making at least $2^{\Omega(m)}\tau/\mathrm{polylog}(M) > 1/\tau$ many queries. This completes the proof of Theorem 4.3. $\square$

## 5 Conclusion and Future Directions

In this work, we showed that no efficient SQ algorithm can approximate the optimal error within any constant factor for learning single neurons with Massart noise. In the process, we constructed new moment-matching distributions corresponding to Boolean halfspaces with Massart noise, which is a result of independent interest. Importantly, our construction has some additional desirable properties which allows us to establish hardness for learning low-weight LTFs, strengthening the result of [DK22]. In addition, we provide a simple technique for transforming our binary construction into hardness of learning real-valued single neurons with Massart noise.

A number of avenues for future work remain, some of which we briefly discuss below. Recent work [DKK$^+$22] studied the problem of learning halfspaces under the Gaussian distribution with Massart noise for $\eta = 1/2$. It is plausible that the $\eta = 1/2$ case in our distribution-independent setting is much harder than the $\eta = 0.49$ case. Establishing such a statement is left as an interesting open question. Another direction concerns the distribution-specific setting. Are there efficient algorithms with non-trivial error guarantees (e.g., achieving a constant factor approximation) for learning single neurons under simple discrete distributions (e.g., under the uniform distribution on the hypercube)?

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
