## Supplementary Material

## A   Omitted Technical Preliminaries

Here we record definitions and facts that will be used in our proofs.

**Definition A.1** (Pairwise Correlation). The pairwise correlation of two distributions with probability mass functions (pmfs) $D_1, D_2 : \{0,1\}^M \to \mathbb{R}_+$ with respect to a distribution with pmf $D : \{0,1\}^M \to \mathbb{R}_+$, where the support of $D$ contains the supports of $D_1$ and $D_2$, is defined as $\chi_D(D_1, D_2) + 1 \stackrel{\text{def}}{=} \sum_{x \in \{0,1\}^M} D_1(x) D_2(x)/D(x)$. We say that a collection of $s$ distributions $\mathcal{D} = \{D_1, \ldots, D_s\}$ over $\{0,1\}^M$ is $(\gamma, \beta)$-correlated relative to a distribution $D$ if $|\chi_D(D_i, D_j)| \le \gamma$ for all $i \ne j$, and $|\chi_D(D_i, D_j)| \le \beta$ for $i = j$.

The following notion of dimension effectively characterizes the difficulty of the decision problem.

**Definition A.2** (SQ Dimension). For $\gamma, \beta > 0$, a decision problem $\mathcal{B}(\mathcal{D}, D)$, where $D$ is fixed and $\mathcal{D}$ is a family of distributions over $\{0,1\}^M$, let $s$ be the maximum integer such that there exists $\mathcal{D}_D \subseteq \mathcal{D}$ such that $\mathcal{D}_D$ is $(\gamma, \beta)$-correlated relative to $D$ and $|\mathcal{D}_D| \ge s$. We define the *Statistical Query dimension* with pairwise correlations $(\gamma, \beta)$ of $\mathcal{B}$ to be $s$ and denote it by $\mathrm{SD}(\mathcal{B}, \gamma, \beta)$.

The connection between SQ dimension and lower bounds is captured by the following lemma.

**Lemma A.3** ([FGR+17]). *Let $\mathcal{B}(\mathcal{D}, D)$ be a decision problem, where $D$ is the reference distribution and $\mathcal{D}$ is a class of distributions over $\{0,1\}^M$. For $\gamma, \beta > 0$, let $s = \mathrm{SD}(\mathcal{B}, \gamma, \beta)$. Any SQ algorithm that solves $\mathcal{B}$ with probability at least $2/3$ requires at least $s \cdot \gamma/\beta$ queries to the $\mathrm{STAT}(\sqrt{2\gamma})$ oracles.*

We have the following fact about the chi-squared inner product in the discrete setting.

**Fact A.4.** *For distributions $\mathbf{P}, \mathbf{Q}$ over $\{0,1\}^M$, we have that $1 + \chi_{U_M}(\mathbf{P}, \mathbf{Q}) = \sum_{T \subseteq [M]} \widehat{\mathbf{P}}(T) \widehat{\mathbf{Q}}(T)$.*

We will also use the following standard fact:

**Fact A.5.** *Let $m, M \in \mathbb{Z}_+$ with $m < M$. For any constant $0 < c < 1$ and $M > 2m/c$, there exists a collection $\mathcal{C}$ of $2^{\Omega_c(m)}$ subsets $S \subseteq [M]$ such that any pair $S, S' \in \mathcal{C}$, with $S \ne S'$, satisfies $|S \cap S'| < cm$.*

In fact, an appropriate size set of random subsets satisfies the above statement with high probability.

The following correlation lemma states that the distributions $\mathbf{P}_S^A$ are nearly orthogonal as long as $A$ satisfies the nearly moment-matching condition.

**Lemma A.6** (Correlation Lemma [DKS22]). *Let $k, m, M \in \mathbb{Z}_+$ with $k \le m \le M$. If the distribution $A$ on $[m] \cup \{0\}$ satisfies Condition 3.3, then for all $S, S' \subseteq [M]$ with $|S| = |S'| = m$, we have that*

$$|\chi_{U_M}(\mathbf{P}_S^A, \mathbf{P}_{S'}^A)| \le (|S \cap S'|/m)^{k+1} \chi^2(A, \mathrm{Bin}(m, 1/2)) + k\nu^2 . \tag{1}$$

## B   Omitted Proofs from Section 3

### B.1   Proof of Proposition 3.5

Let $\mathcal{C}$ be a collection of $s = 2^{\Omega(m)}$ subsets $S \subseteq [M]$ with $|S| = m$ whose pairwise intersections are all less than $m/2$. By Fact A.5 (taking the local parameter $c = 1/2$), such a set is guaranteed to exist. We then need to show that for $S, S' \in \mathcal{C}$, we have that $|\chi_{U_M^p}(\mathbf{P}_{S,a,b}^{A,B,p}, \mathbf{P}_{S',a,b}^{A,B,p})|$ is small. Since $U_M^p, \mathbf{P}_{S,a,b}^{A,B,p}$, and $\mathbf{P}_{S',a,b}^{A,B,p}$ all assign $y = 1$ with probability $p$, it is not hard to see that

$$\chi_{U_M^p}(\mathbf{P}_{S,a,b}^{A,B,p}, \mathbf{P}_{S',a,b}^{A,B,p}) = p \, \chi_{U_M^p|y=1}\left((\mathbf{P}_{S,a,b}^{A,B,p} \mid y = 1), (\mathbf{P}_{S',a,b}^{A,B,p} \mid y = 1)\right) +$$

$$(1-p) \, \chi_{U_M^p|y=-1}\left((\mathbf{P}_{S,a,b}^{A,B,p} \mid y = -1), (\mathbf{P}_{S',a,b}^{A,B,p} \mid y = -1)\right)$$

$$= p \, \chi_{U_M}(\mathbf{P}_S^A, \mathbf{P}_{S'}^A) + (1-p) \, \chi_{U_M}(\mathbf{P}_S^B, \mathbf{P}_{S'}^B).$$

By Lemma A.6, for $S, S' \in \mathcal{C}$ with $S \neq S'$, it holds that

$$\chi_{U_M^p}(\mathbf{P}_{S,a,b}^{A,B,p}, \mathbf{P}_{S',a,b}^{A,B,p}) \leq k\nu^2 + 2^{-k}(\chi^2(A, \text{Bin}(m, 1/2)) + \chi^2(B, \text{Bin}(m, 1/2))) \leq \tau \ .$$

If $S = S'$, a similar computation shows that

$$\chi_{U_M^p}(\mathbf{P}_{S,a,b}^{A,B,p}, \mathbf{P}_{S,a,b}^{A,B,p}) = \chi^2(\mathbf{P}_{S,a,b}^{A,B,p}, U_M^p) \leq \chi^2(A, \text{Bin}(m, 1/2)) + \chi^2(B, \text{Bin}(m, 1/2)) \ .$$

Let $\gamma = \tau$ and $\beta = \chi^2(A, \text{Bin}(m, 1/2)) + \chi^2(B, \text{Bin}(m, 1/2))$. We have that the Statistical Query dimension of this testing problem with correlations $(\gamma, \beta)$ is at least $s$. Then applying Lemma A.3 with $(\gamma, \beta)$ completes the proof.

## B.2 Proof of Lemma 3.8

The conditions on $\mu$ define a linear program (LP). We will show that this LP is feasible by showing that the dual LP is infeasible. The dual LP asks for a degree at most $k$ real polynomial $q(x)$ such that

$$|q(0)| \geq (1/11) \sum_{i=1-s}^{s-1} |q(i)| \ .$$

Consider the parameterization $p(\theta) = q(s \sin(\theta))$. We will leverage the fact that $p(\theta)$ is a degree-$k$ polynomial in $e^{i\theta}$ and $e^{-i\theta}$. In particular, $p(\theta)$ can be written as

$$p(\theta) = \sum_{j=-k}^{k} a_j e^{ij\theta} \ ,$$

for some complex coefficients $a_j \in \mathbb{C}$. By normalizing, we can assume that $\sum_{j=-k}^{k} |a_j|^2 = 1$. Then, for any $\theta$, we have that

$$|p(\theta)| \leq \sum_{j=-k}^{k} |a_j| = O(\sqrt{k}) \ ,$$

where the final inequality follows from the Cauchy-Schwarz. In particular, $|q(0)| = |p(0)| = O(\sqrt{k})$. In addition, for any $\theta$, by Cauchy-Schwarz, we have that

$$|p'(\theta)| = \left| \sum_{j=-k}^{k} j a_j e^{ij\theta} \right| \leq \sum_{j=-k}^{k} |j||a_j| \leq \sqrt{\sum_{j=-k}^{k} j^2} = O(k^{3/2}) \ .$$

Finally, we note that

$$\frac{1}{2\pi} \int_0^{2\pi} |p(\theta)|^2 d\theta = \sum_{j=-k}^{k} |a_j|^2 = 1 \ .$$

Combining the latter with the fact that $|p(\theta)| = O(\sqrt{k})$, we obtain that

$$\int_0^{2\pi} |p(\theta)| d\theta = \Omega(k^{-1/2}) \ .$$

For any $\theta \in [0, 2\pi]$, let $n(\theta)$ be the closest $\phi \in [0, 2\pi]$ such that $s \sin(\phi)$ is an integer in $\{1 - s, 2 - s, \ldots, s - 1\}$. It is not hard to see that $|n(\theta) - \theta| = O(s^{-1/2})$ for all such $\theta$. Furthermore, we have that

$$|p(n(\theta)) - p(\theta)| \leq |n(\theta) - \theta| \sup_{\theta' \in [0, 2\pi]} |p'(\theta')| \leq O(k^{3/2} s^{-1/2}) \ .$$

We can thus write

$$\Omega(k^{-1/2}) = \int_0^{2\pi} |p(\theta)| d\theta \leq \int_0^{2\pi} |p(n(\theta))| d\theta + O(k^{3/2} s^{-1/2}) \ .$$

Therefore,

$$\int_0^{2\pi} |p(n(\theta))| d\theta \geq \Omega(k^{-1/2}) \ .$$

On the other hand, each value of $p(n(\theta))$ is equal to the value of $q$ evaluated at some integer between $1 - s$ and $s - 1$. Furthermore, it is not hard to see that each such integer occurs for at most a total of $O(s^{-1/2})$ range of $\theta$'s. Therefore, we get that

$$O(s^{-1/2}) \sum_{i=1-s}^{s-1} |q(i)| \geq \Omega(k^{-1/2}) \ .$$

Combining with the fact that $|q(0)| = O(k^{1/2})$, this shows that it is impossible that

$$|q(0)| \geq 1/4 \sum_{i=1-s}^{s-1} |q(i)| \ .$$

This completes our proof.

## C  Omitted Proofs from Section 4

### C.1  Proof of Claim 4.2

For a $\mathbf{v}_S$ the vector whose $i^{th}$ coordinate is 1 if $i \in S$ and 0 otherwise, let $g : \{0, 1\}^{m'} \to \{\pm 1\}$ be defined as $g(\mathbf{x}) = -1$ if and only if $\mathbf{v}_S^T \mathbf{x} \in J$. In this way, we are able to write $g$ as a degree-$2d$ PTF, i.e., $g(\mathbf{x}) = \mathrm{sign}(\prod_{z \in J}(\mathbf{v}_S^T \mathbf{x} - z)^2)$. Therefore, there exists some LTF $L : \mathbb{R}^M \to \{\pm 1\}$ such that $g(\mathbf{x}) = L(\mathbf{x}') = L(V_{2d}(\mathbf{x}))$ for all $\mathbf{x}$. We now bound the error for LTF $L$ under the distribution $(\mathbf{X}', Y')$. By the law of total probability, we have that

$$\mathbf{Pr}_{(\mathbf{X}', Y')}[Y' \neq L(\mathbf{X}')] = \mathbf{Pr}_{(\mathbf{X}, Y)}[Y \neq g(\mathbf{X})]$$
$$\leq \mathbf{Pr}_{(\mathbf{X}, Y)}[Y \neq g(\mathbf{X}) \mid Y = 1] + \mathbf{Pr}_{(\mathbf{X}, Y)}[Y \neq g(\mathbf{X}) \mid Y = -1] \ .$$

We note that our hard distribution returns $(\mathbf{x}', y')$ with $y' = L(\mathbf{x}')$, unless it picked a sample corresponding to a sample of $\mathcal{D}_-$ coming from $\overline{J}$, therefore,

$$\mathbf{Pr}_{(\mathbf{X}', Y')}[Y' \neq L(\mathbf{X}')] \leq \mathbf{Pr}_{(\mathbf{X}, Y)}[Y \neq g(\mathbf{X}) \mid Y = -1] \leq \zeta \ ,$$

which implies that $\mathrm{OPT}_{\mathrm{Mass}} \leq \zeta \leq \exp(-\Omega(\log(M)^{8/9}))$. We then show that $(\mathbf{X}', Y')$ is a Massart LTF distribution with noise rate upper bound of $\eta = 1/3$. For any fixed $\mathbf{x}' \in \mathbb{R}^M$, we have that

$$\frac{\mathbf{Pr}_{(\mathbf{X}', Y')}[Y' = 1 \mid \mathbf{X}' = \mathbf{x}']}{\mathbf{Pr}_{(\mathbf{X}', Y')}[Y' = -1 \mid \mathbf{X}' = \mathbf{x}']} = \frac{\mathbf{Pr}_{(\mathbf{X}, Y)}[Y = 1 \mid \mathbf{X} = \mathbf{x}]}{\mathbf{Pr}_{(\mathbf{X}, Y)}[Y = -1 \mid \mathbf{X} = \mathbf{x}]}$$
$$= \frac{\mathbf{Pr}_{(\mathbf{X}, Y)}[Y = 1] \cdot \mathbf{Pr}_{(\mathbf{X}, Y)}[\mathbf{X} = \mathbf{x} \mid Y = 1]}{\mathbf{Pr}_{(\mathbf{X}, Y)}[Y = -1] \cdot \mathbf{Pr}_{(\mathbf{X}, Y)}[\mathbf{X} = \mathbf{x} \mid Y = -1]} = \frac{\|\mathcal{D}_+\|_1 \cdot \mathbf{P}_S^{\mathcal{D}_+}(\mathbf{x})}{\|\mathcal{D}_-\|_1 \cdot \mathbf{P}_S^{\mathcal{D}_-}(\mathbf{x})} = \frac{\mathcal{D}_+(\mathbf{v}_S^T \mathbf{x})}{\mathcal{D}_-(\mathbf{v}_S^T \mathbf{x})} \ .$$

Therefore, if $\mathbf{v}_S^T \mathbf{x} \in J$, the above ratio will be 0 and $L(\mathbf{x}') = -1$, which means that the noise rate $\eta(\mathbf{x}') = 0$; otherwise the above ratio will be at least 2 (since $\mathcal{D}_+ > 2\mathcal{D}_-$ on $\overline{J}$ by property 1(b) of Proposition 3.6) and $L(\mathbf{x}') = 1$, which means that $\eta(\mathbf{x}') \leq 1/3$. This completes the proof of the claim.

### C.2  Proof of Claim 4.5

Let $\mathbf{v}_S$ be the vector whose $i^{th}$ coordinate is 1 if $i \in S$ and 0 otherwise. By Lemma 4.4, there is a real univariate polynomial $p$ of degree $O(d)$ such that $p(\mathbf{v}_S^T \mathbf{x}) = 1, \mathbf{v}_S^T \mathbf{x} \in J$ and $p(\mathbf{v}_S^T \mathbf{x}) \leq 0, \mathbf{v}_S^T \mathbf{x} \notin J$. Let $g(\mathbf{x}) := \widehat{\mathrm{ReLU}}(p(\mathbf{v}_S^T \mathbf{x}))$. Since the absolute value of every coefficient of $p$ is at most $m^{O(d)} = \mathrm{poly}(M)$, by our definition, the total weight of the corresponding neuron $g$ is at most $m^{O(d)} = \mathrm{poly}(M)$. Therefore, there exists some $\widehat{\mathrm{ReLU}}$ function $L : \mathbb{R}^M \to \mathbb{R}$ such

that $g(\mathbf{x}) = L(\mathbf{x}') = L(V_{O(d)}(\mathbf{x}))$ for all $\mathbf{x}$. We now bound the error for $L$ under the distribution $(\mathbf{X}', Y')$. By the law of total expectation, we have that

$$\mathbf{E}_{(\mathbf{X}', Y')} \left[ (Y' - L(\mathbf{X}'))^2 \right] = \mathbf{E}_{(\mathbf{X}, Y)} \left[ (Y - g(\mathbf{X}))^2 \right]$$
$$\leq \mathbf{E}_{(\mathbf{X}, Y)} \left[ (Y - g(\mathbf{X}))^2 \mid Y = 1 \right] + \mathbf{E}_{(\mathbf{X}, Y)} \left[ (Y - g(\mathbf{X}))^2 \mid Y = -1 \right] .$$

We note that our hard distribution returns $(\mathbf{X}', Y')$ with $Y' = L(\mathbf{X}')$, unless it picked a sample corresponding to a sample of $\mathcal{D}_-$ coming from $\bar{J}$, therefore,

$$\mathbf{E}_{(\mathbf{X}', Y')} \left[ (Y' - L(\mathbf{X}'))^2 \right] \leq \mathbf{E}_{(\mathbf{X}, Y)} \left[ (Y - g(\mathbf{X}))^2 \mid Y = 1 \right] \leq 4\zeta .$$

which implies that $\mathrm{OPT}_{\mathrm{Mass}} \leq 4\zeta \leq \exp(-\Omega(\log(M)^{8/9}))$. We then show that $(\mathbf{X}', Y')$ is a Massart single neuron distribution with $\widehat{\mathrm{ReLU}}$ activation and with noise rate upper bound of $\eta = 1/3$. For any fixed $\mathbf{x}' \in \mathbb{R}^M$, we have that

$$\frac{\mathbf{Pr}_{(\mathbf{X}', Y')}[Y' = -1 \mid \mathbf{X}' = \mathbf{x}']}{\mathbf{Pr}_{(\mathbf{X}', Y')}[Y' = 1 \mid \mathbf{X}' = \mathbf{x}']} = \frac{\mathbf{Pr}_{(\mathbf{X}, Y)}[Y = -1 \mid \mathbf{X} = \mathbf{x}]}{\mathbf{Pr}_{(\mathbf{X}, Y)}[Y = 1 \mid \mathbf{X} = \mathbf{x}]}$$
$$= \frac{\mathbf{Pr}_{(\mathbf{X}, Y)}[Y = -1] \cdot \mathbf{Pr}_{(\mathbf{X}, Y)}[\mathbf{X} = \mathbf{x} \mid Y = -1]}{\mathbf{Pr}_{(\mathbf{X}, Y)}[Y = 1] \cdot \mathbf{Pr}_{(\mathbf{X}, Y)}[\mathbf{X} = \mathbf{x} \mid Y = 1]} = \frac{\|\mathcal{D}_+\|_1 \cdot \mathbf{P}_S^{\mathcal{D}_+}(\mathbf{x})}{\|\mathcal{D}_-\|_1 \cdot \mathbf{P}_S^{\mathcal{D}_-}(\mathbf{x})} = \frac{\mathcal{D}_+(\mathbf{v}_S^T \mathbf{x})}{\mathcal{D}_-(\mathbf{v}_S^T \mathbf{x})} .$$

Therefore, if $\mathbf{v}_S^T \mathbf{x} \in J$, the above ratio will be 0 and $L(\mathbf{x}') = -1$, which means that the noise rate $\eta(\mathbf{x}') = 0$; otherwise the above ratio will be at least 2 (since $\mathcal{D}_+ > 2\mathcal{D}_-$ on $\bar{J}$ by property 1(b) of Proposition 3.6) and $L(\mathbf{x}') = 1$, which means that $\eta(\mathbf{x}') \leq 1/3$. This completes the proof of the claim.

# D   SQ Hardness of Learning a Single Neuron with $L_2$-Massart Noise

In this section, we prove our SQ hardness result of learning a single neuron with fast convergent activations and $L_2$-Massart noise. Without loss of generality, we consider activations which converge on the negative side. For such an activation $f$, let $f_- := f(-\infty)$ and $c_+$ be a constant such that $f(c_+) \neq f_-$. The main theorem of this section is the following.

**Theorem D.1** (SQ Hardness of $L_2$-Massart Learning). *Let $f : \mathbb{R} \to \mathbb{R}$ be a fast convergent activation. Any SQ algorithm that learns a single neuron with activation $f$ on $\mathbb{R}^M$, in the presence of $\eta$-$L_2$-Massart noise with $\eta = \frac{2(f(c_+) - f_-)^2}{9}$, to squared error better than $1/\mathrm{poly}(\log(M))$ requires either queries of accuracy better than $\tau := \exp(-\Omega(\log(M)^{1.05}))$ or at least $1/\tau$ statistical queries. This holds even if:*

1. *The optimal neuron has squared error $\mathrm{OPT}_{\mathrm{Mass-L2}} \leq \exp(-\Omega(\log(M)^{8/9}))$,*

2. *The $\mathbf{X}$ values are supported on $\{0, 1\}^M$, and*

3. *The total weight of the neuron is $\mathrm{poly}(M)$.*

*Proof.* Our proof will make use of the SQ framework of Section 3.1 and will crucially rely on the one-dimensional construction of Proposition 3.6. In this section, we fix the labels $a = f_-, b = f(c_+)$, and apply the construction in Section 3.3 to obtain the joint distributions $(\mathbf{X}, Y)$ and $(\mathbf{X}', Y')$. Note that $y = y'$ and there is a known 1-1 mapping between $\mathbf{x}$ and $\mathbf{x}'$, therefore finding a hypothesis that predicts $y'$ given $\mathbf{x}'$ is equivalent to finding a hypothesis for $y$ given $\mathbf{x}$.

**Claim D.2.** *The distribution $(\mathbf{X}', Y')$ on $\{0, 1\}^M \times \{f_-, f(c_+)\}$ is an $L_2$-Massart single neuron distribution with respect to activation $f$, it has optimal squared error $\mathrm{OPT}_{\mathrm{Mass-L2}} \leq \exp(-\Omega(\log(M)^{8/9}))$ and $L_2$-Massart noise rate upper bound of $\eta = \frac{2(f(c_+) - f_-)^2}{9}$.*

*Proof.* We assume $M > |c_+|$ to be sufficiently large. Let $\mathbf{v}_S$ be the vector whose $i^{th}$ coordinate is 1 if $i \in S$ and 0 otherwise. By Lemma 4.4, there is a real univariate polynomial $q(x)$ of degree $O(d)$ such that $q(x) = 1, \forall x \in J$ and $q(x) \leq 0, \forall x \in \bar{J}$. Let $p(x) = (c_+ + M)q(x) - M$ and $g(\mathbf{x}) = f(p(\mathbf{v}_S^T \mathbf{x}))$. By definition, we have that $p(x) = c_+$ for $x \in J$ and $p(x) \leq -M$ for $x \in \bar{J}$.

Since the absolute value of every coefficient of $p$ is at most $m^{O(d)} = \text{poly}(M)$, the weight of the corresponding neuron $g$ is at most $m^{O(d)} = \text{poly}(M)$. Therefore, there exists some fast convergent activation $L : \mathbb{R}^M \to \mathbb{R}$ such that $g(\mathbf{x}) = L(\mathbf{x}') = L(V_{O(d)}(\mathbf{x}))$ for all $\mathbf{x}$. We now bound the error for $L$ under the distribution $(\mathbf{X}', Y')$. We note that conditional on $Y = f_-$, we will always have that $\mathbf{v}_S^T \mathbf{x} \notin J$ and conditional on $Y = f(c_+)$, we will have that $\mathbf{v}_S^T \mathbf{x} \notin J$ with probability at most $\zeta$. Therefore, by the law of total expectation, we have that

$$\mathbf{E}_{(\mathbf{X}', Y')}[(Y' - L(\mathbf{X}))^2] = \mathbf{E}_{(\mathbf{X}, Y)}[(Y - g(\mathbf{X}))^2]$$

$$\leq \mathbf{E}_{(\mathbf{X}, Y)}[(Y - g(\mathbf{X}))^2 \mid Y = f_-] + \mathbf{E}_{(\mathbf{X}, Y)}[(Y - g(\mathbf{X}))^2 \mid Y = f(c_+)]$$

$$\leq \mathbf{E}_{(\mathbf{X}, Y)}[(f_- - g(\mathbf{X}))^2 \mid Y = f_-] + 2\zeta \mathbf{E}_{(\mathbf{X}, Y)}[(f_- - f(c_+))^2 + (f_- - g(\mathbf{X}))^2 \mid \mathbf{v}_S^T \mathbf{X} \notin J, Y = f(c_+)]$$

$$\leq 1/\text{poly}(M) + 2\zeta \cdot (1/\text{poly}(M) + (f_- - f(c_+))^2)$$

$$\leq \exp(-\Omega(\log(M)^{8/9})) + \exp(-\Omega(\log(M)^{8/9})) \cdot (1/\text{poly}(M) + (f_- - f(c_+))^2)$$

$$\leq \exp(-\Omega(\log(M)^{8/9})) ,$$

where the third inequality follows from the definition of fast convergent activation. Therefore, we have that $\text{OPT}_{\text{Mass}-L2} \leq \exp(-\Omega(\log(M)^{8/9}))$. We then show that $(\mathbf{X}', Y')$ is a $L_2$-Massart single neuron distribution with activation $f$ and with noise rate upper bound of $\eta = \frac{2(f(c_+) - f_-)^2}{9}$. Note that for any $\mathbf{x} \in \mathbb{R}^{m'}$, if $\mathbf{v}_S^T \mathbf{x} \in J$, then $g(\mathbf{x}) = f(p(\mathbf{v}_S^T \mathbf{x})) = f(c_+)$ and $Y$ will always be $f(c_+)$, which implies that the error will always be 0. Hence, we assume that $\mathbf{v}_S^T \mathbf{x} \notin J$ and have that

$$\frac{\mathbf{Pr}_{(\mathbf{X}, Y)}[Y = f_- \mid \mathbf{X} = \mathbf{x}]}{\mathbf{Pr}_{(\mathbf{X}, Y)}[Y = f(c_+) \mid \mathbf{X} = \mathbf{x}]} = \frac{\mathbf{Pr}_{(\mathbf{X}, Y)}[Y = f_-] \cdot \mathbf{Pr}_{(\mathbf{X}, Y)}[\mathbf{X} = \mathbf{x} \mid Y = f_-]}{\mathbf{Pr}_{(\mathbf{X}, Y)}[Y = f(c_+)] \cdot \mathbf{Pr}_{(\mathbf{X}, Y)}[\mathbf{X} = \mathbf{x} \mid Y = f(c_+)]}$$

$$= \frac{\|\mathcal{D}_+\|_1 \cdot \mathbf{P}_S^{\mathcal{D}_+}(\mathbf{x})}{\|\mathcal{D}_-\|_1 \cdot \mathbf{P}_S^{\mathcal{D}_-}(\mathbf{x})} = \frac{\mathcal{D}_+(\mathbf{v}_S^T \mathbf{x})}{\mathcal{D}_-(\mathbf{v}_S^T \mathbf{x})} \geq 2 ,$$

which implies that $\mathbf{Pr}_{(\mathbf{X}, Y)}[Y = f(c_+) \mid \mathbf{X} = \mathbf{x}] \leq 1/3$. Therefore,

$$\mathbf{E}_{(\mathbf{X}', Y')}[(Y' - L(\mathbf{X}'))^2 \mid \mathbf{X}' = \mathbf{x}'] = \mathbf{E}_{(\mathbf{X}, Y)}[(Y - g(\mathbf{X}))^2 \mid \mathbf{X} = \mathbf{x}]$$

$$= (f(c_+) - g(\mathbf{x}))^2 \mathbf{Pr}_{(\mathbf{X}, Y)}[Y = f(c_+) \mid \mathbf{X} = \mathbf{x}] + (f_- - g(\mathbf{x}))^2 \mathbf{Pr}_{(\mathbf{X}, Y)}[Y = f_- \mid \mathbf{X} = \mathbf{x}]$$

$$\leq \frac{(f(c_+) - g(\mathbf{x}))^2}{3} + (f_- - g(\mathbf{x}))^2 \leq \frac{2\left((f(c_+) - f_-)^2 + (f_- - g(\mathbf{x}))^2\right)}{3} + (f_- - g(\mathbf{x}))^2$$

$$\leq \frac{2(f(c_+) - f_-)^2}{3} + 1/\text{poly}(M) \leq \frac{8(f(c_+) - f_-)^2}{9} ,$$

where the third inequality follows from $\mathbf{v}_S^T \mathbf{x} \notin J$ and the definition of fast convergent activation. This completes the proof of the claim. $\qquad\square$

We now show that the $(\mathcal{D}_+, \mathcal{D}_-, f_-, f(c_+), m')$-Hidden Junta Testing Problem efficiently reduces to our learning task. In more detail, we show that any SQ algorithm that computes a hypothesis $h'$ satisfying $\mathbf{E}_{(\mathbf{X}', Y')}[(h'(\mathbf{X}') - Y')^2] < p(1-p)(f_- - f(c_+))^2 - 2\sqrt{2\tau}$ can be used as a black-box to distinguish between $\mathbf{P}_{S,a,b}^{\mathcal{D}_+, \mathcal{D}_-, p}$, for some unknown subset $S \subseteq [m']$ with $|S| = m$, and $U_{m'}^p$. Since there is a 1-1 mapping between $\mathbf{x} \in \{0,1\}^{m'}$ and $\mathbf{x}' \in \{0,1\}^M$, we denote $h : \{0,1\}^{m'} \mapsto \mathbb{R}$ to be $h(\mathbf{x}) = h'(\mathbf{x}')$. We note that we can (with one additional query to estimate the $\mathbf{E}[(h'(\mathbf{X}') - Y')^2]$ within error $\sqrt{2\tau}$) distinguish between (i) the distribution $\mathbf{P}_{S,a,b}^{\mathcal{D}_+, \mathcal{D}_-, p}$, and (ii) the distribution $U_{m'}^p$. This is because for any $h$ we have that

$$\mathbf{E}_{(\mathbf{X}, Y) \sim U_{m'}^p}[(h(\mathbf{X}) - Y)^2] = \mathbf{E}_{(\mathbf{X}, Y) \sim U_{m'}^p}[h(\mathbf{X})^2] - 2\mathbf{E}_{(\mathbf{X}, Y) \sim U_{m'}^p}[h(\mathbf{X})]\mathbf{E}_{(\mathbf{X}, Y) \sim U_{m'}^p}[Y]$$

$$+ \mathbf{E}_{(\mathbf{X}, Y) \sim U_{m'}^p}[Y^2]$$

$$\geq \mathbf{E}_{(\mathbf{X}, Y) \sim U_{m'}^p}[h(\mathbf{X})]^2 - 2\mathbf{E}_{(\mathbf{X}, Y) \sim U_{m'}^p}[h(\mathbf{X})]\mathbf{E}_{(\mathbf{X}, Y) \sim U_{m'}^p}[Y]$$

$$+ \mathbf{E}_{(\mathbf{X}, Y) \sim U_{m'}^p}[Y^2]$$

$$\geq \mathbf{E}_{(\mathbf{X}, Y) \sim U_{m'}^p}[Y^2] - \mathbf{E}_{(\mathbf{X}, Y) \sim U_{m'}^p}[Y]^2 = p(1-p)(f_- - f(c_+))^2.$$

Applying Proposition 3.5, we determine that any SQ algorithm which, given access to a distribution $\mathbf{P}$ so that either $\mathbf{P} = U_{m'}^p$, or $\mathbf{P}$ is given by $\mathbf{P}_{S,a,b}^{\mathcal{D}_+,\mathcal{D}_-,p}$ for some unknown subset $S \subseteq [m']$ with $|S| = m$, correctly distinguishes between these two cases with probability at least $2/3$ must either make queries of accuracy better than $\sqrt{2\tau}$ or must make at least $2^{\Omega(m)}\tau/(\chi^2(A, \mathrm{Bin}(m, 1/2)) + \chi^2(B, \mathrm{Bin}(m, 1/2)))$ statistical queries. Therefore, it is impossible for an SQ algorithm to learn a hypothesis with error better than $p(1-p)(f_- - f(c_+))^2 - 2\sqrt{2\tau} = \Theta(1/s) - \Theta(\sqrt{\tau}) = 1/\mathrm{polylog}(M)$ without either using queries of accuracy better than $\tau$ or making at least $2^{\Omega(m)}\tau/\mathrm{polylog}(M) > 1/\tau$ many queries. This completes the proof of Theorem D.1. $\qquad\square$