# OpenReview forum: "SQ Lower Bounds for Learning Single Neurons with Massart Noise"
_NeurIPS.cc/2022/Conference — NeurIPS 2022 Accept_

### Official Review · Reviewer_DKp1 · 2022-06-23

**Rating:** 5
**Confidence:** 4
**Soundness:** 2 fair
**Presentation:** 2 fair
**Contribution:** 3 good

**Summary:**

The authors consider the problem of (distribution-free) PAC learning of single neuron neural nets in the presence of Massart noise, with the aim off finding a constant factor approximation error. They prove a super polynomial lower bound in the statistical query model (a restricted family of algorithms). This is proved for ReLU and also a broader class of activation function: those that output zero for negative parts and any injective function for the positive part. They establish a hardness gap that is super-constant and scales with the dimensionality of the problem. This lower bound shows a barrier to improve the existing upper bounds (i.e., one cannot hope to use algorithms based on statistical queries to get an efficient method with constant approximation guarantee)

Technical overview:
•	The proof starts by finding a SQ hard instance for the Boolean case based on recent results for continuous case.
•	The recent result is for continuous case and finds instances for one-dimensional Gaussian like distributions and extend it to high dimensions.
•	They need the case for discrete and it is not easy to extend that Gaussian like results to this case
•	They start by constructing the binomial distribution conditioned on X being 0 mod s and then change the distributions so that (1) conditioned on X being far from n/2 the probability of Y=1 is larger than Y=-1 (2) conditioned on Y=1 and Y=-1, the distributions have less overlap and (3) the conditional distributions have moments close to binomial.
•	Use recent result to extend this to high dimensions.
•	Construct an instance of ReLU that produces the desired classifier for the Boolean case


**Questions:**

N/A

**Strengths And Weaknesses:**

Strengths:

+ The authors offer a SQ lower bound for learning neural nets with 1 neurons under Massart noise. This is a solid technical contribution and an important step towards understanding the barriers for improving regression algorithms under noise conditions.

Weaknesses:

+ The generalization of the Massart noise model to the real-valued case (Definition 1) seems somewhat unnatural. In the real valued extension the y-values are kept "exactly the same" with some probability and changed otherwise. Yes, this coincides with Massart noise for binary case, but in the continuous case is rather odd. Some more motivations can help.

+ The paper does not provide enough background and motivations, has missing/vague notations, missing explanations that can glue different technical results, and sometimes not rigorous in the statements of the results. Overall, the paper is hard to read or assess, and in my opinion is not ready for publication. The presentation can be greatly improved. Some concrete points:

+ Many notations are unclear, vague, or even undefined. Here are some examples:
+  Definition 3.2. The notation (X_i )_(i∈S) is confusing and has not been properly defined or explained. Also, what is a symmetric distribution in this case? Parsing this definition with these notations
+  Definition 3.4. What does sampling from (P_S^A,a) mean? Does it mean that we sample a point x from P_S^A   and return (x,a)? The notation is quite confusing.
+ Proposition 3.5. although rather standard, it would help to define the X^2 distribution and its notation.
+ When you use a measure as a function, do you assume it has some kind of (unnormalized) density?
+ Related to the previous point, the notations D_+ and D_- are not consistently used. Sometimes they refer to the function as used D_+=0 on J and sometimes they are used with reference to the input as D_+ (x). Also, it would help to define the ell_1 norm of D in this context.
+ would help to explicitly define (±1)-weight half-space

+ The introduction of the paper does not motivate the setting. Adding some more motivations such as explaining why a statistical query lower bound is important can help the non-expert reader. The structure of the literature review and the background does not help the non-expert reader either.
+ Some lemmas and propositions appear in the main paper without enough context or explanations about where they are going to be used.
+ The technical proofs are quite hard to follow. The different parts of the proofs are not introduced with intuitive introductions and explanations. It is difficult for the reader to obtain a general understanding of the problem without diving into the technical proofs of theorems, which are also hard to understand considering the missing notations and definitions.

---

> ### Author Response · Authors · 2022-08-02
> **Response to Reviewer DKp1**
>
> We thank the reviewer for reading our results and their detailed comments. We respond to the concerns in the following. We ask that the reviewer consider increasing the score, given that we have addressed their concerns.
>
> 1. *"Many notations are unclear, vague, or even undefined. When you use a measure as a function, do you assume it has some kind of (unnormalized) density?"*
>
> We thank the reviewer for pointing this issue out. We will fix the notation issues and improve the presentation in the revision. Indeed, all measures $D_+,D_-,\mathcal{R}$ are unnormalized measures in Section 3.2, and the "expectation" and "moments" concepts here are defined over the unnormalized measure. We apologize that the conclusion of Lemma 3.7 loses an factor “$s$” in the statement since $\| D_- \|_1 = \Theta(1/s)$ and Condition 3.3 is phrased in terms of exact “distributions”, i.e., $\nu=s\binom{m}{k}\exp(-\Omega(m/s^2))$. However, this does not affect our parameter setting for constructions in Section 3.3 and our SQ hardness results in the later sections, since this missing factor "$s$" is not the leading term.
>
> The notation in Definition 3.2 and Definition 3.4 are fairly standard in this line of work. Specifically: (X_i)\_(i∈S) clearly denotes the marginal distribution over the coordinates in the subset $S$. For the distribution P\_(S,a,b)^(A,B,p), with probability $p$, we draw a random sample $\mathbf{x}$ from $P_S^A$ and return $(\mathbf{x},a)$; with probability $1-p$, we draw a random sample $\mathbf{x}$ from $P_S^B$ and return $(\mathbf{x},b)$. In addition, we define our "pairwise correlation" $\chi^2(\cdot,\cdot)$ in the appendix due to page limit.
>
>
> 2. *"The generalization of the Massart noise model to the real-valued case (Definition 1) seems somewhat unnatural. In the real valued extension the y-values are kept "exactly the same" with some probability and changed otherwise. Yes, this coincides with Massart noise for binary case, but in the continuous case is rather odd."*
>
> As we explained in our common response, our "Massart Noise Model" (Definition 1.1) is a natural generalization of the classic Massart noise for binary classification problem, that has been studied in several prior works (including [Diakonikolas-Park-Tzamos, NeurIPS’21], [Chen-Koehler-Moitra-Yau, FOCS’21] and [Chen-Koehler-Moitra-Yau, STOC’22]). In particular, our result answers an open problem proposed in [Diakonikolas-Park-Tzamos, NeurIPS’21].
>
> If the reviewer’s concern about Definition 1.1 is that it is relatively weak (as the adversarial must leave the value of $y$ *exactly* unchanged the majority of the time): This should not be considered as a disadvantage of our work, as it makes our result stronger for showing a *lower bound* with this limited noise.
>
> In addition, we also introduce a different "$L_2$-Massart Noise Model" (Definition 1.3) in which the label noise has bounded variance and our techniques work for this "$L_2$-Massart Model'' as well. In some scenarios, Definition 1.3 can be considered as more realistic than Definition 1.1.
>
> References:
>
> [DPT21] Diakonikolas, I., Park, J. H., & Tzamos, C. (2021). ReLU Regression with Massart Noise. Advances in Neural Information Processing Systems, 34, 25891-25903.
>
> [CKMY21] Chen, S., Koehler, F., Moitra, A., & Yau, M. (2022, February). Online and distribution-free robustness: Regression and contextual bandits with huber contamination. In 2021 IEEE 62nd Annual Symposium on Foundations of Computer Science (FOCS) (pp. 684-695). IEEE.
>
> [CKMY22] Chen, S., Koehler, F., Moitra, A., & Yau, M. (2022, June). Kalman filtering with adversarial corruptions. In Proceedings of the 54th Annual ACM SIGACT Symposium on Theory of Computing (pp. 832-845).

---

> > ### Comment · Reviewer_DKp1 · 2022-08-08
> > **thanks for the clarifications**
> >
> > >If the reviewer’s concern about Definition 1.1 is that it is relatively weak (as the adversarial must leave the value of  exactly unchanged the majority of the time): This should not be considered as a disadvantage of our work, as it makes our result stronger for showing a lower bound with this limited noise.
> >
> > That's a fair point. I increased my score to 5.
> >
> > In absence of the presentation issues I would have given the paper a higher score. I think the paper would be much more appreciated if the issues with notations and rigor, (lack of) background and motivations, and overall the flow information are addressed in the future.

---

### Official Review · Reviewer_J7z6 · 2022-07-08

**Rating:** 5
**Confidence:** 4
**Soundness:** 4 excellent
**Presentation:** 2 fair
**Contribution:** 2 fair

**Summary:**

This paper studies the question of distribution-free, improper learning of single neurons when a random fraction of the labels in the training data have been corrupted. The objective is to output a predictor whose squared loss over the clean data is as close to that achieved by the optimal neuron. They show that no statistical query algorithm with polynomial query complexity or inverse polynomial tolerance can always achieve a constant factor approximation to the optimal squared loss in the worst case. They additionally show hardness in a different setting that they call "L_2-Massart noise" in which the labels are given by perturbing the true label by heterogeneous noise of bounded variance.

They also show a new SQ lower bound for distribution-free learning of halfspaces in the presence of Massart noise. Unlike the previous lower bounds of Chen et al. '20 and Diakonikolas and Kane '20, their bound applies to halfspaces over the hypercube whose coefficients are from {1, -1}.

The hard instance they construct to prove these lower bounds closely follows the approach of Diakonikolas and Kane '20, with the main technical twist that they need to handle discrete rather than Gaussian inputs. They design a joint distribution over $(x,y)$ such that 1) conditioned on $y = -1$ or on $y = 1$, the distribution over $x$ is iid random bits outside of a hidden subset of $m$ coordinates over which it is symmetric and such that the 1D distribution of the sum of these $m$ coordinates approximately moment-matches $\text{Bin}(m,1/2)$, 2) these 1D distributions for $y = 1$ and $y = -1$ don't overlap by much, and 3) the joint distribution can be realized as an instance of whatever learning problem their lower bound applies to.

**Questions:**

*Questions*:
- A priori isn't it possible that ReLU regression under Massart noise isn't constant-factor-approximable even when the covariates are *uniform* over the hypercube?
- Can one hope for a common generalization of this work together with Diakonikolas and Kane '20 to show hardness for a more general family of activations than just {sign} + {"relu-like" functions that are injective over the positive part}? Perhaps any activation that is far from the identity function?

*Minor comments*:
- Technical overview is quite hard to read for a general audience as it assumes close familiarity with [DK20]. Also it uses notation that is not defined previously in the paper (see below)
- P. 2 Line 58-60: While this is true for binary classification, this point doesn't seem very relevant for the real-valued regression problems studied in this submission.
- P. 4 Line 147: The relevant document doesn't appear to be available, though the associated talk slides which are available online don't seem to mention this as an open problem
- P. 4 Line 168: s not defined
- P. 4 Line 171: n not defined
- P. 4 Line 152: LTF and PTF should be defined
- P. 9: would be helpful to include a sentence on why it's more convenient to work with $\widehat{\text{ReLU}}$ rather than $\text{ReLU}$
- Lemma 3.7: confusing as Condition 3.3 is phrased in terms of distributions, whereas Lemma 3.7 is about an unnormalized measure, and this is exacerbated by the expectation notation used in the proof. also, isn't $\mathcal{R}$ also unnormalized? if so, the use of the word "pmf" is similarly confusing.
- Some of the technical writing is rather hard to read, e.g. the block of text in Section 3.3, though this might be due to space constraints.
- It seems more accurate to call Definition 1.1 Huber contamination rather than Massart noise.

**Limitations:**

The authors give a pretty comprehensive answer to the question they set out to solve, so I don't see any limitations per se. I also don't see any potential negative societal impact, as they are proving a hardness result for a theoretical problem.

**Strengths And Weaknesses:**

The topic of this paper feels a bit like smashing a bunch of well-studied topics together. Arguably the main reason to be interested in extending existing lower bounds for learning halfspaces under Massart noise to learning ReLUs would be that real-valued regression problems tend to require new techniques for showing hardness. But the architecture of the proof in this work builds heavily on Diakonikolas and Kane '20, and the hard joint distribution exhibited ultimately has binary labels as before (they've gone from the PTFs of Diakonikolas and Kane '20 to certain ReLUs of polynomials which take on binary values over the hypercube). The main technical difference is that it's important that the x's are over the hypercube instead of $\mathbb{R}^d$, so one needs some discrete analogue of the usual non-Gaussian component analysis setup. But this discrete analogue is precisely the one developed in Diakonikolas, Kane, and Sun '22. That being said, the actual moment-matching construction is new and requires a number of clever moves, in particular Lemma 3.8, that are specific to the setting at hand.

---

> ### Author Response · Authors · 2022-08-02
> **Response to Reviewer J7z6**
>
> We thank the reviewer for reading our results and their detailed comments. We respond to the questions and concerns in the following. We ask that the reviewer consider increasing the score, given that we have addressed their comments and concerns.
>
> We start by addressing the concerns within the review in order:
>
> 1. *"The topic of this paper feels a bit like smashing a bunch of well-studied topics together. Arguably the main reason to be interested in extending existing lower bounds for learning halfspaces under Massart noise to learning $\mathrm{ReLUs}$ would be that real-valued regression problems tend to require new techniques for showing hardness. But the architecture of the proof in this work builds heavily on Diakonikolas and Kane '20."*
>
> We emphasize that our results are not achieved by simply applying the framework or combining and extending existing results in [DK20] and [DKS22]. While the generic SQ lower bound construction we use is a supervised learning analogue of [DKS22], the main challenge is to construct new moment matching distributions corresponding to Boolean halfspaces with Massart noise. In our paper, we provide a novel univariate moment matching construction for discrete distributions, which is our main technical contribution (Proposition 3.6, Lemma 3.7 and Lemma 3.8).
>
> We would like to note that there have been several COLT/NeurIPS papers in the past few years that all work by applying the (same) generic SQ lower bound construction, developed in [DKS17], based on Non-Gaussian Component Analysis. Each of these papers need to develop novel univariate moment matching distributions, which is their main technical contribution.
>
> We clearly state in our paper that the generic discrete SQ lower bound construction is a supervised learning analogue of [DKS22]; and that the main technical challenge and novelty is the construction of our univariate moment matching distributions. In addition, our work contains novel ideas for transforming the classification hardness into the hardness of learning single (real-valued) neurons.
>
> Although our work borrows some high-level ideas from [DK20], our univariate moment-matching for discrete distributions is novel and technically deep. Importantly, our construction has some additional desired properties which allows us to give hardness for learning low-weight LTFs, which is stronger than the result in [DK20]. In addition, we provide a novel technique for transforming our binary construction into hardness of learning real-valued single neurons with Massart noise. Note that if one tries to apply this technique on the old construction of [DK20], it would not work, since the positive labels will not have a single fixed value, which makes the labels’ value leaking extra information to the learner.
>
> 2. *"The hard joint distribution exhibited ultimately has binary labels as before (they've gone from the PTFs of Diakonikolas and Kane '20 to certain $\mathrm{ReLUs}$ of polynomials which take on binary values over the hypercube)."*
>
> The fact that our construction has binary labels should not be considered as a disadvantage. The instance, by itself, is a valid hard instance for learning real-valued $\mathrm{ReLUs}$ with Massart noise; any algorithm for this learning task cannot avoid this hard instance.
>
> Additionally, reducing to binary instances is almost necessary in order to prove SQ lower bounds (rather than CSQ or other weaker bounds) using the known framework for producing such bounds. This is because proving such bounds requires finding distributions $(\mathbf{x},y)$ that have finite $\chi^2$-norm with respect to a given base distribution. As the noiseless distributions cannot have full support in $\mathbb{R}^{M+1}$ (as we can have at most one y-value for each $\mathbf{x}$), these lower bounds will usually only be possible if the base distribution usually has only a small number of possible y-values for each $\mathbf{x}$, with binary functions being one of the most convenient instantiations of this.
>
> 3. *"Notation Clarification in Lemma 3.7 and the proof."*
>
> We thank the reviewer for pointing out this issue. Indeed, all measures $D_+,D_-,\mathcal{R}$ are unnormalized measures here, and the "expectation" and "moments" concepts in the proof are defined over the unnormalized measure. We apologize that the conclusion of Lemma 3.7 loses a factor "$s$" in the statement, since $\|D_-\|_1=\Theta(1/s)$ and Condition 3.3 is phrased in terms of exact "distributions'', i.e., $\nu=s\binom{m}{k}\exp(-\Omega(m/s^2))$. However, this does not affect our choice of the parameter setting for the constructions in Section 3.3 and our SQ hardness results in the later sections, since this missing factor "$s$" is not the leading term. We will fix other notation issues in the revision as well.

---

> > ### Author Response · Authors · 2022-08-02
> > **Response to Reviewer J7z6 (continuing)**
> >
> > 4. *"Would be helpful to include a sentence on why it's more convenient to work with $\mathrm{\widehat{ReLU}}$ rather than $\mathrm{ReLU}$."*
> >
> > As we mentioned in the paper, this is due to technical convenience (Lemma 4.4). In particular, we need the activation function taking value $(-1,+1)$ instead of $(0,1)$ to make the labels consistent with our notation $D_+$ and $D_-$.
> >
> > 5. *"It seems more accurate to call Definition 1.1 Huber contamination rather than Massart noise."*
> >
> > If we consider a Huber contaminated joint distribution of $(\mathbf{x},y)$, the adversary has the power to also change the marginal distribution of $\mathbf{x}$. This is a substantially more challenging noise model than ours, where only the labels can be perturbed. Our definitions (both Definition 1.1 and Definition 1.3) are more benign semi-random noise models.
> >
> > We address the reviewer’s questions as follows:
> >
> > 1. *"A priori isn't it possible that $\mathrm{ReLU}$ regression under Massart noise isn't constant-factor-approximable even when the covariates are uniform over the hypercube?"*
> >
> > We are not sure whether there is a polynomial time algorithm that achieves constant-factor approximation, even for the special case where the covariates are uniform over the hypercube. To our best knowledge, the recent work [Diakonikolas-Park-Tzamos, NeurIPS’21] provides a computationally efficient algorithm for learning $\mathrm{ReLU}$ regression under Massart noise that achieves exact parameter recovery with high probability in the model under mild anti-concentration assumptions on the covariates.
> >
> > Furthermore, we do not believe that it is known whether the easier problem of learning LTFs is constant-factor approximable with uniform covariates and Massart noise.
> >
> >
> > 2. *"Can one hope for a common generalization of this work together with Diakonikolas and Kane '20 to show hardness for a more general family of activations than just {sign} + {"relu-like" functions that are injective over the positive part}? Perhaps any activation that is far from the identity function?"*
> >
> > We apologize about the misleading statement we have on lines 99-102 where we claim that the activation function needs to be injective on the positive side. This is in fact not necessary. Although Theorem 1.2 is stated only for ReLU, it holds for any function with a "threshold behavior" as well, namely, for an activation function $f$, for which there are absolute constants $c$ and $c'$ such that for any $t<c, f(t)=c'$. For Theorem 1.5, we establish SQ hardness results for any activation function which converges sufficiently fast on the negative side (Definition 1.4) under the different "$L_2$-Massart Noise" model (Definition 1.3).
> >
> > The construction in [DK20] is insufficient for the purpose of proving SQ lower bounds for learning single neurons with Massart noise. As we mentioned earlier, if one tries to apply this technique on the old construction of [DK20], it would not work, since the positive labels will not have a single fixed value, which makes the labels’ value leaking extra information to the learner.
> >
> > Furthermore, while the full strength of our threshold condition may not be necessary, some kind of condition is necessary on the activation function. For example, if $f$ is the identity function, we have linear regression, which *can* be solved even in the presence of noise with $L_1$ or $L_2$-regression (depending on the noise model). Furthermore, if $f$ is a low degree polynomial, the problem can be solved by polynomial regression. Finally, if $f$ is injective, then the samples $(\mathbf{x},f^{-1}(y))$ fit a linear model and (depending on the noise model) linear regression can again be used. Our thresholding condition is a simple condition that rules out all of these examples, and while there may well be weaker conditions that suffice, one would need to be careful to find one that still rules out the above examples where efficient regression is possible.
> >
> > References:
> >
> > [DKS17] Diakonikolas, I., Kane, D. M., & Stewart, A. (2017, October). Statistical query lower bounds for robust estimation of high-dimensional gaussians and gaussian mixtures. In 2017 IEEE 58th Annual Symposium on Foundations of Computer Science (FOCS) (pp. 73-84). IEEE.
> >
> > [DK20] Diakonikolas, I., & Kane, D. (2022, June). Near-optimal statistical query hardness of learning halfspaces with massart noise. arXiv preprint arXiv:2012.09720
> >
> > [DPT21] Diakonikolas, I., Park, J. H., & Tzamos, C. (2021). ReLU Regression with Massart Noise. Advances in Neural Information Processing Systems, 34, 25891-25903.
> >
> > [DKS22] Diakonikolas, I., Kane, D. M., & Sun, Y. (2022). Optimal SQ Lower Bounds for Robustly Learning Discrete Product Distributions and Ising Models. arXiv preprint arXiv:2206.04589.

---

> > > ### Comment · Reviewer_J7z6 · 2022-08-08
> > > **Thanks for the clarifications**
> > >
> > > I'm happy to raise my score to a 5: I think 4 was too harsh given that, as I acknowledged in my review, the moment-matching constructions are very clever and nontrivial. Additionally, the authors clearly clarify in the response the ways in which the techniques are not just a combination of [DK20] and [DKS22]. Also, the responses to questions 4 onwards were very illuminating.
> > >
> > > That said-- and this might still be overly harsh and a matter of taste-- but I still stand by my original reservation that the *topic* of this paper needs better motivation. I think I can get on board with *agnostic* ReLU regression over nice distributions; via Frank-Wolfe, this is connected to learning more sophisticated function classes like depth two networks. I think learning *halfspaces* under Massart noise was important to study because in practice it's reasonable to see how, e.g., kernel methods for classification can be made robust to semirandom noise. Is there any downstream reason (practical or theoretical) to study Massart noise for ReLU regression, beyond the fact that it's in between agnostic and realizable ReLU regression?

---

> > > > ### Author Response · Authors · 2022-08-08
> > > > **Significance of Topic/Results**
> > > >
> > > > We agree with the reviewer that, in principle, the significance of a research work can be considered a matter of personal taste.
> > > > In this instance, however, we would like to make the following general points:
> > > >
> > > > 1) Historically speaking, the study of PAC learning (in the context of both boolean-valued and real-valued functions) started in the distribution-independent setting. Distribution-specific PAC learning became a topic of interest and investigation much later -- once researchers realized that there are regimes in which distribution-independent learning is either too challenging or intractable. That is, in a host of settings, what we would have *really* liked to efficiently solve is distribution-independent learning. If we cannot do that, we impose assumptions on the distribution on examples (e.g., log-concavity) and *hope* that these assumptions are justifiable in our settings of interest. (This remark applies orthogonally from the existence of noise/corruption in the training data.)
> > > >
> > > > 2) Closer to the context of the current paper, it is of fundamental importance to understand how the distributional assumptions and the choice of the noise model interact to affect the complexity of learning natural concept classes. While this question is fairly well-understood for the agnostic model, very little had been known until recently for semi-random noise models and in particular for the Massart noise model(s) we study in this paper. Specifically, it was known that *distribution-independent agnostic* learning of simple concept classes (including halfspaces and ReLUs) was really hard (even to approximate); but it was in principle possible that distribution-independent *Massart* learning for such concept classes has non-trivial algorithms.
> > > >
> > > > 3) This was confirmed for the class of halfspaces in the award winning algorithmic work of Diakonikolas-Gouleakis-Tzamos (best paper at NeurIPS'19); and in a host of subsequent works, including (Chen et al. FOCS'21) for the case of linear regression and (Diakonikolas-Park-Tzamos NeurIPS'21) for the case ReLU regression. Our SQ lower bounds imply that there may be computational limits in what can be efficiently learned in these semi-random regimes as well (thus answering an open question in the latter work of DPT'21).
> > > >
> > > > 4) Space limitations aside, we chose to not provide additional motivation (in the introduction of our paper)
> > > > on ReLU regression (and related problems) because *these problems are not new*. They have been studied (in the distribution-independent setting) by prior *algorithmic* works (including the ones mentioned above). We refer the reviewer to these prior works that provide a number of settings where such noise can appear and the significance of the distribution-independent setting.
> > > >
> > > > 5) We did not fully understand the point behind the last comment made by the reviewer (regarding agnostic ReLU regression under Gaussian and the Franke-Wolfe algorithm). The NeurIPS'20 paper by Goel-Gollakota-Klivans comes to mind. Our point of view is that noise-tolerant learning of simple (linear or non-linear) functions is not necessarily a tool in the context of learning deeper neural networks, but of interest in its own right. Related to this, as explained in our original response, our SQ-hardness is not restricted to ReLUs, but applies to a range of other activations.
> > > >
> > > > We are happy to discuss further with the reviewer regarding their thoughts on the matter.

---

### Official Review · Reviewer_8CnK · 2022-07-09

**Rating:** 7
**Confidence:** 3
**Soundness:** 4 excellent
**Presentation:** 4 excellent
**Contribution:** 4 excellent

**Summary:**

This work deals with the problem of PAC learning a single neuron with Massart noise. For this task for various activation functions, the authors provide super-polynomial SQ lower bounds. In particular, it is proved that no efficient SQ algorithm can approximate the optimal error within any constant factor.

**Questions:**

Theorem 1.2 captures the ReLU activation function and other functions with a "threshold" behavior. Is there some intuition on why the result holds only for these activations? Do we expect similar hardness results for other activations?

**Limitations:**

This is a theoretical work. The authors describe the limitations of this work.

**Strengths And Weaknesses:**

The main question of the paper is whether there exists an efficient constant approximate learning algorithm for distribution-independent learning of a single neuron with Massart noise for various activation functions. The authors rule out the existence of efficient SQ algorithms with constant factor approximation for the above problem for the ReLU activation.

Next the authors introduce a second variant of the Massart noise model for real-valued functions and give another SQ lower bound for this model and a general family of activations (including ReLUs).

I believe that the most interesting result is Theorem 1.6 which gives an SQ hardness result for low-weight Massart halfspaces on the Boolean hypercube. The fact that this problem remains SQ-hard is quite nice and justifies the difficulties when dealing with the Massart noise model.

In general, the paper is clear and the results are well presented.

---

> ### Author Response · Authors · 2022-08-02
> **Response to Reviewer 8CnK**
>
> We would like to thank the reviewer for their effort and positive assessment of our work. We respond to the question below:
>
> *"Theorem 1.2 captures the ReLU activation function and other functions with a "threshold" behavior. Is there some intuition on why the result holds only for these activations? Do we expect similar hardness results for other activations?"*
>
> The threshold behavior can be defined as follows: for an activation function $f$, there are constants $c$ and $c'$, such that for any $t < c, f(t) = c'$. Importantly, we do not need the function to be injective on the positive side. (We apologize about the misleading statement we made in lines 99-102, which says the function has to be injective on the positive side. This is not necessary.) Our proof for Theorem 1.2 can be used to show hardness for any activation function satisfying this "threshold" behavior.
>
> An intuition is that for a sample $(\mathbf{x},y)$ such that $\langle \mathbf{w},\mathbf{x} \rangle < c, y = f(\langle \mathbf{w}, \mathbf{x} \rangle) = c'$ is fixed. Therefore, upon seeing $y$, the only information the learner knows is $\langle \mathbf{w}, \mathbf{x}\rangle < c$, and has no idea about how small $\langle \mathbf{w}, \mathbf{x}\rangle$ is. This allows us to hide a lot information from the learner and make this argument: for those $(\mathbf{x},y)$ that $\mathbf{x} \sim P_S^{D_+}$, $(\mathbf{x},y)$ has just as much information as $\mathbf{x}$ itself (since $y$ is always $c’$), then we can argue learning any information from these $\mathbf{x}$ is as hard as distinguishing $\mathbf{x}\sim P_S^{D_+}$ versus $\mathbf{x}\sim U_M$.
>
> In addition, it is not hard to see that this restriction cannot be readily removed without invalidating our hardness results. For example, if the activation function is the identity (i.e., $f(\mathbf{x}) = \mathbf{x}$, and $y = \langle \mathbf{w} , \mathbf{x} \rangle$), then the problem *can* be solved efficiently using either least-squares or $L_1$-regression, depending on the noise model. Similarly, if $f$ is any injective function, then (before noise) $f^{-1}(y)$ will be linear in $\mathbf{x}$ and so for at least some noise models (e.g., Definition 1.1), there will be efficient algorithms. That being said, the threshold behavior that we use is a rather strict way to ensure that we can hide information from the algorithm, and it might be worth investigating whether it can be replaced by weaker assumptions.

---

### Official Review · Reviewer_yXsm · 2022-07-11

**Rating:** 7
**Confidence:** 2
**Soundness:** 4 excellent
**Presentation:** 3 good
**Contribution:** 3 good

**Summary:**

The paper addresses the problem of PAC learning a single neuron under the Massart noise model. For various activation functions, including ReLUs, the authors show that no efficient Statistical Query (SQ) algorithm can approximate the optimal error within any constant factor. The key technical innovation is a new SQ-hardness construction for learning neurons with a sign activation on the Boolean hypercube.


**Questions:**

Can the case $\eta=1/2$ in the noise model be addressed within this framework?

**Limitations:**

I do not foresee any potential negative societal impact of this work.

**Strengths And Weaknesses:**

The paper is well-written and the results coherently presented. The paper can benefit from a separate conclusion / potential open problems section.

---

> ### Author Response · Authors · 2022-08-02
> **Response to Reviewer yXsm**
>
> We would like to thank the reviewer for their effort and positive assessment of our work. We will add a separate conclusion / potential open problems section in the revision. We respond to the question below:
>
> *"Can the case $\eta=1/2$ in the noise model be addressed within this framework?"*
>
> Our Definition 1.1 requires $\eta<1/2$; however, allowing $\eta$ to equal $1/2$ would be a sensible extension of our definition (of course, unless we bound the probability that $\eta(x)=1/2$ above, learning with bounded error will be impossible in many cases). For Definition 1.3, $\eta$ can be $1/2$, but $\eta=1/2$ has no special meaning in that case.
>
> In terms of Definition 1.1, $\eta=1/2$ is the special case that is hardest to give an algorithm and easiest for the purpose of establishing lower bounds. In terms of algorithms, previous work [Diakonikolas-Park-Tzamos, NeurIPS’21] did not consider the case $\eta=1/2$. However, there are works (e.g., [Diakonikolas-Kane-Kontonis-Tzamos-Zarifis, STOC’22]) considering $\eta=1/2$ under the classic binary Massart noise. In terms of lower bounds, any instance that satisfies the $1/3$-massart condition must also satisfy the $1/2$-massart condition. So our SQ-hardness result (Theorem 1.2) would immediately apply to the $\eta=1/2$ case.
>
> Although our work does not focus on this case, it is plausible that the $\eta=1/2$ case is much harder than the $\eta = 0.49$ case (as [Diakonikolas-Kane-Kontonis-Tzamos-Zarifis, STOC’22] show holds in certain binary classification models). This would be an interesting avenue for further investigation.
>
> References:
>
> [DPT21] Diakonikolas, I., Park, J. H., & Tzamos, C. (2021). ReLU Regression with Massart Noise. Advances in Neural Information Processing Systems, 34, 25891-25903.
>
> [DKKTZ22] Diakonikolas, I., Kane, D. M., Kontonis, V., Tzamos, C., & Zarifis, N. (2022, June). Learning general halfspaces with general massart noise under the gaussian distribution. In Proceedings of the 54th Annual ACM SIGACT Symposium on Theory of Computing (pp. 874-885).

---

### Author Response · Authors · 2022-08-02
**Overall Response**

We thank the reviewers for the detailed feedback. We are encouraged by the positive scores from reviewers *yXsm* and *8CnK*. We will address the questions and concerns by responding directly to each reviewer, and here we wish to highlight the following key points about our work.

We emphasize that our results are not achieved by simply applying the framework or combining and extending existing results in [DK20] and [DKS22]. While the generic SQ lower bound construction we use is a supervised learning analogue of [DKS22], the main challenge is to construct new moment matching distributions corresponding to Boolean halfspaces with Massart noise. In our paper, we provide a novel univariate moment matching construction for discrete distributions, which is our main technical contribution (Proposition 3.6, Lemma 3.7 and Lemma 3.8). We would like to note that there have been several COLT/NeurIPS papers in the past few years that all work by applying the (same) generic SQ lower bound construction, developed in [DKS17], based on Non-Gaussian Component Analysis. Each of these papers need to develop novel univariate moment matching distributions, which is their main technical contribution.

Although our work borrows some high-level ideas from [DK20], our univariate moment-matching for discrete distributions is novel and technically deep. Importantly, our construction has some additional desired properties which allows us to give hardness for learning low-weight LTFs, which is stronger than the result in [DK20]. In addition, we provide a novel technique for transforming our binary construction into hardness of learning real-valued single neurons with Massart noise. Note that if one tries to apply this technique on the old construction of [DK20], it would not work, since the positive labels will not have a single fixed value, which makes the labels’ value leaking extra information to the learner.

We believe our noise models (both Definition 1.1 and Definition 1.3) are natural generalizations of the classic Massart noise for binary classification problems. Our "Massart noise model" (Definition 1.1) for real-valued functions is not new: it is a natural generalization of the classic Massart noise for binary classification problem, and has been studied in several published works (e.g., the same noise model appears in [Diakonikolas-Park-Tzamos, NeurIPS’21], [Chen-Koehler-Moitra-Yau, FOCS’21] and [Chen-Koehler-Moitra-Yau, STOC’22]). In particular, our result answers an open problem posed in [Diakonikolas-Park-Tzamos, NeurIPS’21]. In addition, we also introduce a different "$L_2$-Massart Noise Model" (Definition 1.3) in which the label noise has bounded variance and our techniques work for this "$L_2$-Massart model". In some scenarios, Definition 1.3 can be considered as more suitable than Definition 1.1.

We apologize for the misleading statement made on lines 99-102 (where we claim that the activation function needs to be injective on the positive side). This in fact is not necessary. Although Theorem 1.2 is stated only for $\mathrm{ReLU}$ activations, it holds for any function with a threshold behavio as well; namely, for an activation function $f$, for which there are absolute constants $c$ and $c'$ such that for any $t<c, f(t)=c'$. For Theorem 1.5, we only need the activation function to converge sufficiently fast on the negative side (Definition 1.4).

References:

[DKS17] Diakonikolas, I., Kane, D. M., & Stewart, A. (2017, October). Statistical query lower bounds for robust estimation of high-dimensional gaussians and gaussian mixtures. In 2017 IEEE 58th Annual Symposium on Foundations of Computer Science (FOCS) (pp. 73-84). IEEE.

[DK20] Diakonikolas, I., & Kane, D. (2022, June). Near-optimal statistical query hardness of learning halfspaces with massart noise. arXiv preprint arXiv:2012.09720

[DPT21] Diakonikolas, I., Park, J. H., & Tzamos, C. (2021). ReLU Regression with Massart Noise. Advances in Neural Information Processing Systems, 34, 25891-25903.

[CKMY21] Chen, S., Koehler, F., Moitra, A., & Yau, M. (2022, February). Online and distribution-free robustness: Regression and contextual bandits with huber contamination. In 2021 IEEE 62nd Annual Symposium on Foundations of Computer Science (FOCS) (pp. 684-695). IEEE.

[CKMY22] Chen, S., Koehler, F., Moitra, A., & Yau, M. (2022, June). Kalman filtering with adversarial corruptions. In Proceedings of the 54th Annual ACM SIGACT Symposium on Theory of Computing (pp. 832-845).

[DKS22] Diakonikolas, I., Kane, D. M., & Sun, Y. (2022). Optimal SQ Lower Bounds for Robustly Learning Discrete Product Distributions and Ising Models. arXiv preprint arXiv:2206.04589.

---

### Meta-Review · Area_Chair_fedk · 2022-09-03

**Recommendation:** Accept
**Confidence:** Less certain

**Metareview:**

The paper shows an SQ lower bound for learning a single neuron with a known activation function (functions that includes the rectifier) under Massart noise. The paper makes a sufficiently novel contribution; however, I also agree with the various presentation issues made by the two more critical reviews. The authors must address all of these adequately for the revision

**Award:**

No

---

### Decision · Program_Chairs · 2022-09-14

Accept